# Subunit exchange enhances information retention by CaMKII in dendritic spines

**Dilawar Singh, Upinder Singh Bhalla\***

National Centre for Biological Sciences, Tata Institute of Fundamental Research, Bangalore, India

**Abstract** Molecular bistables are strong candidates for long-term information storage, for example, in synaptic plasticity. Calcium/calmodulin-dependent protein Kinase II (CaMKII) is a highly expressed synaptic protein which has been proposed to form a molecular bistable switch capable of maintaining its state for years despite protein turnover and stochastic noise. It has recently been shown that CaMKII holoenzymes exchange subunits among themselves. Here, we used computational methods to analyze the effect of subunit exchange on the CaMKII pathway in the presence of diffusion in two different micro-environments, the post synaptic density (PSD) and spine cytosol. We show that CaMKII exhibits multiple timescales of activity due to subunit exchange. Further, subunit exchange enhances information retention by CaMKII both by improving the stability of its switching in the PSD, and by slowing the decay of its activity in the spine cytosol. The existence of diverse timescales in the synapse has important theoretical implications for memory storage in networks.

DOI: https://doi.org/10.7554/eLife.41412.001

## Introduction

Memories are believed to be stored in synapses, encoded as changes in synaptic strength (*Hebb, 2005*; *Takeuchi et al., 2014*; *Choi et al., 2018*). Long-term potentiation (LTP), an activity-dependent change in synaptic strength, is considered to be the primary post-synaptic memory mechanism (*Bliss and Collingridge, 2013*; *Mayford et al., 2012*). Various behavioral experiments strongly suggest a critical role for CaMKII in induction of LTP (*Lucchesi et al., 2011*; *Giese et al., 1998*). In the CA1 region of the hippocampus, blocking CaMKII activity blocks the induction of LTP (*Chang et al., 2017*). After LTP induction, several other pathways including protein synthesis (*Aslam et al., 2009*), clustering of receptors (*Shouval, 2005*), receptor translocation (*Hayer and Bhalla, 2005*) and PKM-ζ activation (*Sacktor, 2012*), have been suggested as mechanisms for long-term maintenance of synaptic state. Recent evidence from behavioral assays suggests that CaMKII may also be involved in long-term maintenance of memory (*Rossetti et al., 2017*; but see *Chang et al., 2017*).

Any putative molecular mechanism involved in long-term maintenance of memory must be able to maintain its state despite the potent resetting mechanisms of chemical noise and protein turnover. In the small volume of the synapse (~0.02 μm³ [*Bartol et al., 2015*]), the number of molecules involved in biochemical processes range from single digits to a few hundred, thereby increasing the effect of chemical noise. John Lisman proposed that a kinase and its phosphatase could form a bistable molecular switch able to maintain its state for a very long time despite turnover (*Lisman, 1985*). It has been shown by various mathematical models that CaMKII and its phosphatase protein phosphatase 1 (PP1) may form a bistable switch (*Zhabotinsky, 2000*) which can retain its state for years despite stochastic chemical noise and protein turnover (*Miller et al., 2005*; *Hayer and Bhalla, 2005*). Although there is experimental evidence that CaMKII/PP1 is bistable in in vitro settings (*Bradshaw et al., 2003*; *Urakubo et al., 2014*), experimental evidence for in vivo bistability is

**\*For correspondence:**
bhalla@ncbs.res.in

**eLife digest** The brain stores memories by changing the strength of synapses, the connections between neurons. Synapses that change their strength easily can quickly encode new information. But such synapses are also unstable. They tend to revert back to their original state and so struggle to retain information. By contrast, synapses that are slow to change their strength are slow to learn, but are good at remembering. The difference is a little like that between writing a message in wet sand versus carving it into stone. It is quick and easy to write on sand, but the resulting marks are temporary. Writing on stone is slow and difficult, but the results last far longer.

The brain must strike a balance between how fast synapses can learn and how well they can retain that information. One molecule that helps with this is a synaptic protein called CaMKII. Each CaMKII molecule consists of multiple subunits and exists in either an active or inactive state. Experiments have shown that CaMKII molecules can swap subunits. But how does this affect memory?

Singh and Bhalla used a computer model to simulate subunit exchange between CaMKII molecules. The results revealed that when active CaMKII molecules swap subunits, synapses become better at retaining information. However, when inactive CaMKII molecules swap subunits, synapses do not become better at encoding information. Subunit exchange by CaMKII thus helps synapses stabilize existing memories, rather than form new ones. This makes it easier for the brain to retain stored information despite threats to stability such as the turnover of proteins.

A better knowledge of how the brain balances quick learning and slow forgetting may help us to better understand brain disorders, such as Alzheimer's disease (in which patients struggle to remember), and post-traumatic stress disorder (in which patients struggle to forget). Biological memory networks can also inspire artificial memory systems. Damaging a few components of a computer memory can erase all the stored information. By contrast, the brain loses many neurons every day without suffering the same catastrophic failure. Mimicking such fault tolerance in an artificial system would be highly valuable for storing critical memories.

DOI: https://doi.org/10.7554/eLife.41412.002

lacking. In spine cytosol, CaMKII has been shown not to act like a bistable switch but rather a leaky integrator of calcium activity (*Chang et al., 2017*). However, CaMKII may be bistable in special micro-environments such as the 'core' PSD where it attaches to the NMDA receptor (*Dosemeci et al., 2016*; *Petersen et al., 2003*).

From a computational perspective, the CaMKII/PP1 bistable system is an attractive candidate for memory storage (*Koch, 2004*). Bistability provides a plausible solution to the problem of state maintenance. Previous modeling work has shown that the CaMKII/PP1 system may form a very stable switch despite protein turnover and stochastic noise in the small volume of the synapse. The stability increases exponentially with the number of holoenzymes (*Miller et al., 2005*). It is important to note that this model exhibits bistable behavior only in a narrow range of PP1 concentrations in the PSD. This strict restriction may be met because phosphorylated CaMKII is protected from phosphatases in PSD except PP1 (*Strack et al., 1997a*), which is tightly regulated in the PSD (*Bollen et al., 2010*).

CaMKII has another remarkable property which was hypothesized by Lisman (*Lisman, 1994*) but discovered only recently, namely, subunit exchange. In this process, two CaMKII holoenzymes can exchange active subunits leading to spread of CaMKII activation (*Stratton et al., 2014*).

In this paper, we adapt the Miller and Zhabotinksy (MZ) model (*Miller et al., 2005*) to include subunit exchange and diffusion, and quantify the effects of subunit exchange on the properties of the CaMKII-PP1 system in two adjacent neuronal micro-environments: PSD and spine cytosol.

In the PSD, PP1 is tightly regulated and CaMKII is protected from other phosphatases. But in the spine cytosol, CaMKII is accessible to other phosphatases along with PP1. We examined how state switching lifetimes in the PSD are affected by subunit exchange in different contexts of PP1 levels, turnover, and clustering of CaMKII. In the spine cytosol, we show how the integration of calcium stimuli generates two time-courses of CaMKII activity as a result of subunit exchange (*Chang et al., 2017*).

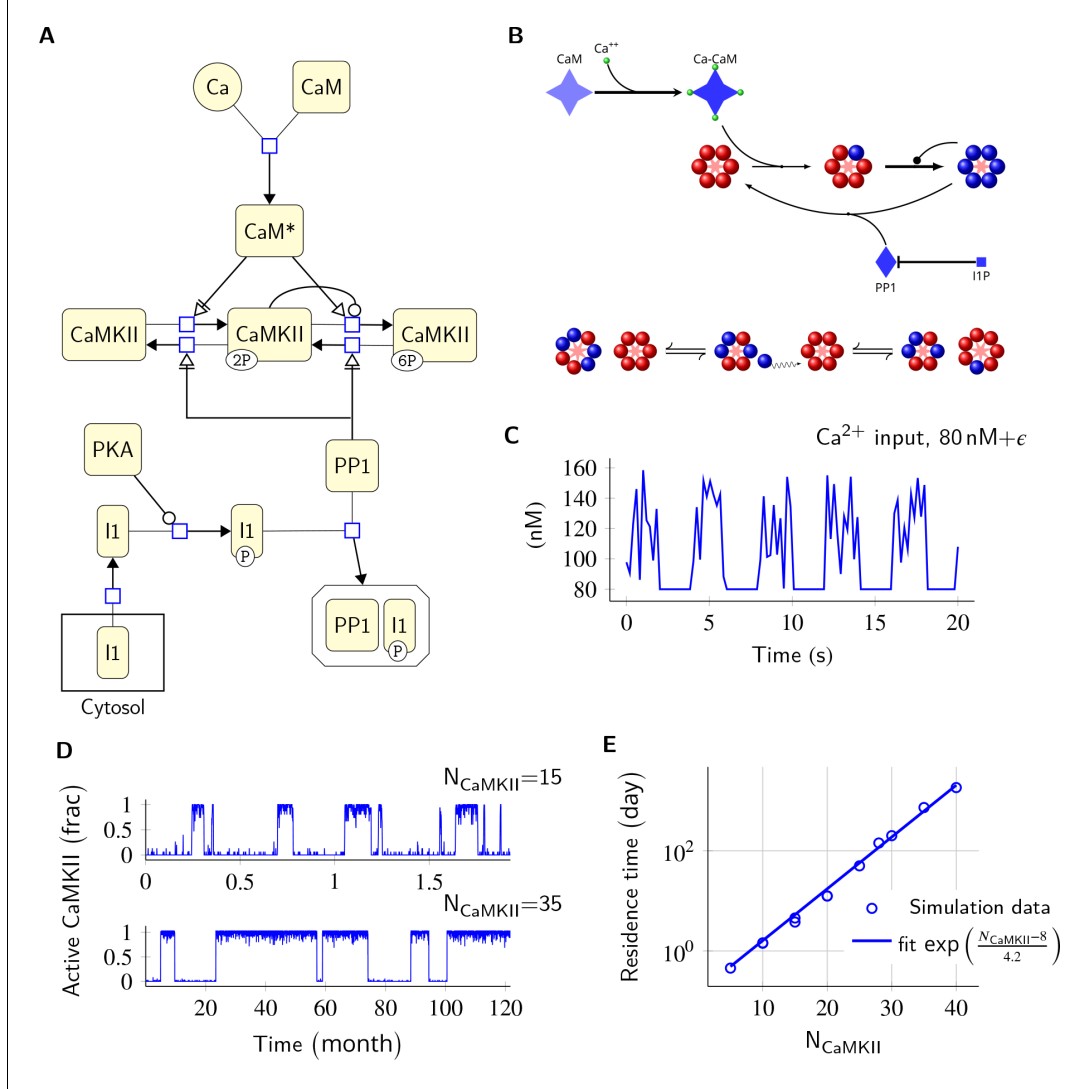

**Figure 1.** Model description and validation. (A) CaMKII/PP1 pathway described in System Biology Graphical Notation (SBGN) – Process Description (PD) Language (*Le Novère et al., 2009*). (B) (above) Major chemical reactions in the CaMKII/PP1 pathway. (below) Subunit exchange between two CaMKII holoenzymes. Blue and red balls represent phosphorylated and un-phosphorylated subunits respectively. (C) Basal $Ca^{2+}$ profile in spine and PSD. Basal $Ca^{2+}$ level is 80 nM with fluctuations every 2 s, lasting for 2 s. These fluctuations (represented by symbol $\epsilon$) are sampled from a uniform distribution with median of 120 nM and range of 40 nM (see Materials and methods). (D) Without diffusion and subunit exchange, CaMKII in our model is bistable. Two trajectories of CaMKII activity (fraction of total CaMKII holoenzymes with at least two subunits phosphorylated) are shown for different system sizes $N_{CaMKII}$ = 15 (top) and $N_{CaMKII}$ = 35 (bottom). (E) Switch stability (measured as average residence time in the stable states) increases exponentially with system size $N_{CaMKII}$. Turnover rate $v_t = 30\,h^{-1}$. Panels C, D, and E show key properties of our model that are very similar to those of the MZ model. Source data are available at https://github.com/dilawar/SinghAndBhalla_CaMKII_SubunitExchange_2018/tree/master/PaperFigures/elifeFigure1 (*Singh and Bhalla, 2018*).

DOI: https://doi.org/10.7554/eLife.41412.003

# Results

## Model validation

The basic computational units in our model are individual CaMKII subunits, and the CaMKII ring consisting of six or seven CaMKII subunits. We treat the CaMKII ring as a proxy for the CaMKII holoenzyme, which consists of two such rings stacked over each other (*Woodgett et al., 1983*; *Hoelz et al., 2003*; *Chao et al., 2011*). We define Active CaMKII as a holoenzyme (ring of six or seven subunits) in which at least two subunits are phosphorylated at Thr[286]. In our model, CaMKII

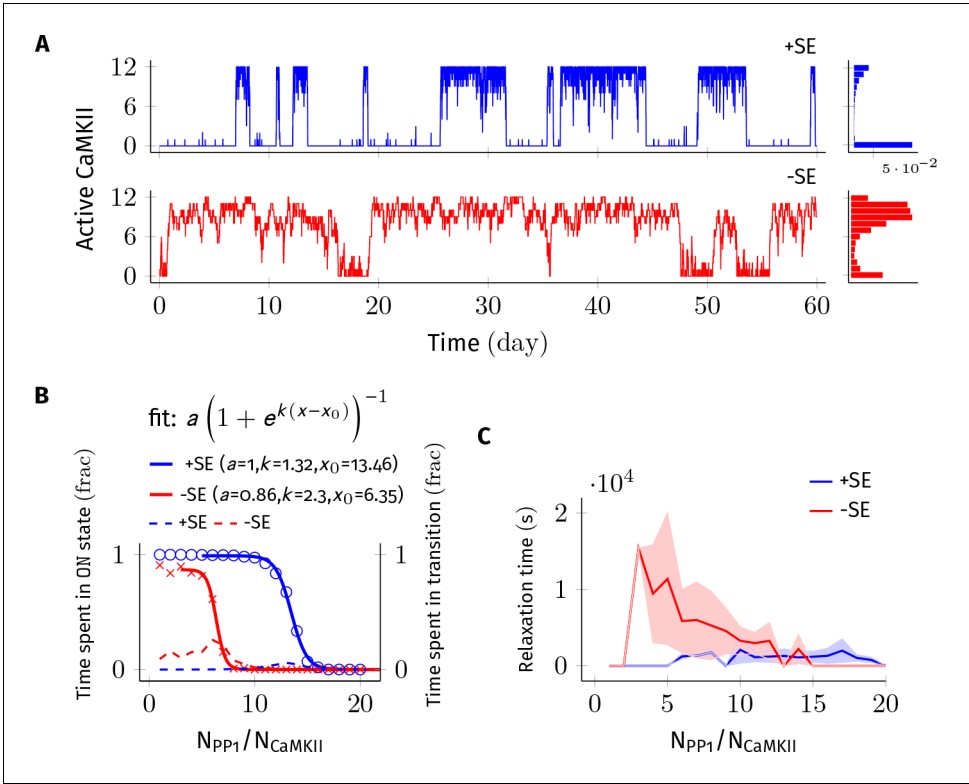

**Figure 2.** Subunit exchange improves the switch's tolerance of PP1 by acting as a compensatory mechanism for the dephosphorylation of CaMKII by PP1. (**A**) Two representative bistable trajectories ($N_{CaMKII}$ = 12) are shown with subunit exchange (+SE, blue) and without subunit exchange (-SE, red) respectively ($D_{sub}$ = 0.1 μm$^2$ s$^{-1}$, and $D_{PP1}$ = 0.5 μm$^2$ s$^{-1}$ for both blue (+SE) and red (-SE); and $N_{PP1}$ = 168 for blue (+SE) and 72 for red (-SE), respectively). (**B**) Blue and red solid lines represent the fraction of total time spend by the switch in the ON state with and without subunit exchange, respectively. The lines are fitted with the function $a/\left(1 + e^{k(x-x_0)}\right)$. Dotted red and blue lines show the fraction of time that the switch spends in intermediate states ($x_a y_{n-a}$, 2 < a < n-2) with and without subunit exchange, respectively. Due to subunit exchange, the switch tolerated a larger amount of PP1 ($x_0$ value 6.35 vs 13.46 that is a change of 7.11×$N_{CaMKII}$). The range of PP1 for which switch remained bistable saw a moderate change ($k$, 1.32 vs. 2.3). The fraction of time spent in intermediate states (dashed lines) is much smaller when subunit exchange is enabled (blue dashed line), that is the switching time is shorter. (**C**) Due to subunit exchange, relaxation time becomes independent of $N_{PP1}$ (blue vs red). Shaded area represents standard deviation. Source data are available at https://github.com/dilawar/SinghAndBhalla_CaMKII_SubunitExchange_2018/tree/master/PaperFigures/elifeFigure2 (*Singh and Bhalla, 2018*).
DOI: https://doi.org/10.7554/eLife.41412.004

exists in 15 possible states compared to two in the MZ model (see Materials and methods). This leads to many more reactions than the MZ model. We also replaced the Michaelis-Menten approximation in the MZ model by equivalent mass-action kinetics (e.g. *Equation 2*). Since analytical comparison of the two models was not possible, we first compared numerical results from our model without diffusion and without subunit exchange with the MZ model (*Figure 1*).

Our model exhibited all the key properties of the MZ model: (1) In the PSD, under basal calcium (Ca$^{2+}$) stimulus conditions, CaMKII/PP1 formed a bistable switch (*Figure 1C,D*), (2) The stability of the switch increased exponentially with system size (*Figure 1E*), (3) Increased number of PP1 molecules ($N_{PP1}$) shut off the switch (*Figure 2*), and (4) Bistability was robust to slow turnover of CaMKII (*Figure 3*).

Thus, our baseline model exhibited all the key properties that had previously been predicted for the bistable CaMKII switch. However, subunit exchange and diffusion introduced several interesting additional properties, which we examine below.

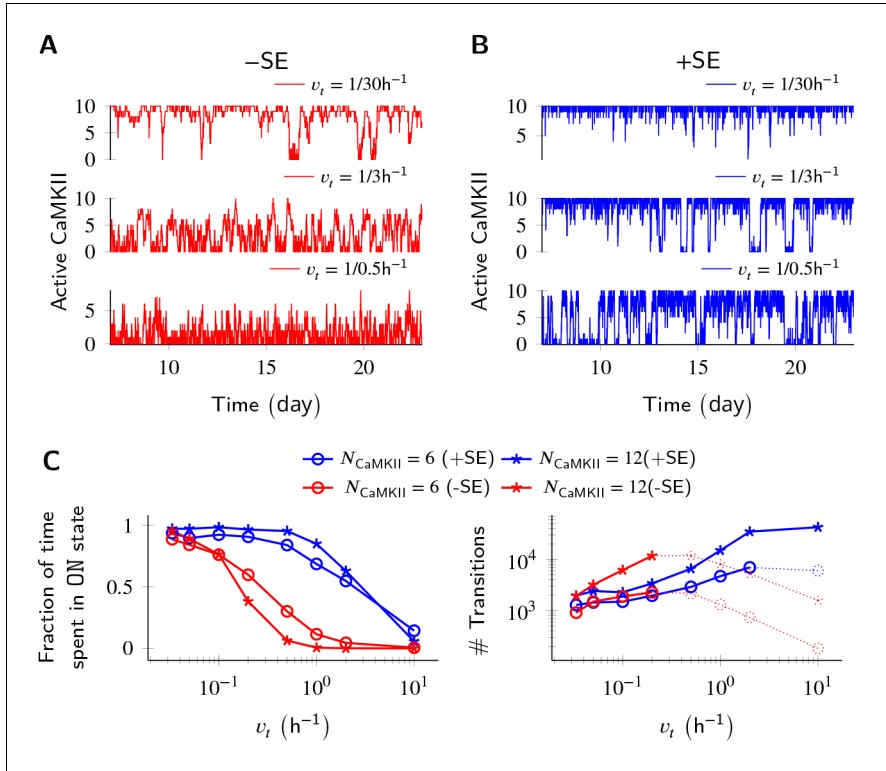

**Figure 3.** Subunit exchange improves switch tolerance of higher rates of protein turnover. (A,B) Three sample trajectories are shown for a switch of size $N_{CaMKII} = 10$ without subunit exchange (-SE, red) and with it (+SE, blue). We consider three different turnover rates of 1 per 30 h, 1 per 3 h, and 1 per 0.5 h. As turnover is increased, the state stability of the ON state of the switch decreases. (C, left) Normalized residence time of the ON state vs. turnover rate for two switches of size 6 and 12. Without subunit exchange, switch stability decreases steeply with turnover rate (red); however, when subunit exchange is enabled, switch stability is not affected by turnover rates as high as 1 h$^{-1}$ (blue). (C, right) In the bistable regime (solid lines), the number of switching events increases monotonically with turnover rate. Source data are available at https://github.com/dilawar/SinghAndBhalla_CaMKII_SubunitExchange_2018/tree/master/PaperFigures/elifeFigure3 (*Singh and Bhalla, 2018*).
DOI: https://doi.org/10.7554/eLife.41412.005

## Subunit exchange increases the tolerance of the CaMKII switch to PP1 and to turnover

We first analyzed the switch sensitivity to PP1. In our model as well in the MZ model, the number of PP1 molecules ($N_{PP1}$) has an upper limit for the switch to exhibit bistability. This constraint arises because PP1 must saturate in the ON state of the switch, that is the maximal enzymatic turnover of PP1 must be smaller than the rate of activation of CaMKII subunits. However, unlike the MZ model where the addition of one extra PP1 molecule changed the residence time of the ON state by roughly 90% (*Figure 2C* in *Miller et al., 2005*), we did not find the residence time of the ON state to be this sensitive to PP1. In our model, on average it required half the number of holoenzymes (i.e. $0.5\times N_{CaMKII}$) extra PP1 molecules to cause a similar 90% change in the residence time of the ON state. This number is roughly equal to the maximum number of CaMKII subunits (released from CaMKII holoenzymes during subunit exchange *Equation 3*) that can exist at any given time in our model. We conjecture that this reduced sensitivity to PP1 is due to the fact that PP1 participates in many more reactions in our model.

We found that a system consisting of $N_{CaMKII} = 12$ holoenzymes remained bistable for $N_{PP1} = 3\times$ to $8\times N_{CaMKII}$ without subunit exchange, and for $N_{PP1} = 12\times$ to $16\times N_{CaMKII}$ with subunit exchange for $D_{sub} = 0.1$, and $D_{PP1} = 0.5$ µm$^2$ s$^{-1}$ (*Figure 2B*). Thus, subunit exchange shifted the middle of the bistable range to higher values of PP1. The width of the range of PP1 over which bistability was

present saw a moderate increase in the presence of subunit exchange (blue and red sigmoidal fit in *Figure 2B*). A similar trend was observed for other values of $D_{PP1}$ and $D_{sub}$ (data not shown).

In the presence of subunit exchange, the ON state of the switch has a tighter distribution (blue vs. red histogram, *Figure 2A*), that is, there are fewer holoenzymes that are completely de-phosphorylated by PP1. We interpret this as follows: In the presence of subunit exchange, any subunit in a holoenzyme de-phosphorylated by the PP1 is likely to be rapidly re-phosphorylated. This is because, when the switch is in ON state, most diffusing subunits present in the PSD are in the phosphorylated state. Hence, in addition to auto-phosphorylation, the exchange reactions (*Equation 3*) turn unphosphorylated holoenzymes to phosphorylated holoenzymes with a significant rate. Taken together, subunit exchange acts as a compensatory mechanism for dephosphorylation by PP1 in the ON state of the switch.

Subunit exchange also had a strong effect on time spent by the switch in transition from one stable state to another (relaxation time). When subunit exchange was enabled, the relaxation time was reduced (red vs. blue dashed line in *Figure 2B*) and also became independent of $N_{PP1}$. As mentioned previously, due to subunit exchange, the ON state has a tighter distribution (blue vs. red histogram in *Figure 2A*). This means that there were fewer ineffective transitions from the ON to the OFF state. As expected, the standard deviation of the relaxation time was also greatly reduced in the presence of subunit exchange (red and blue curve, *Figure 2C*). Thus, subunit exchange makes the switch's ON state less noisy and more robust to dephosphorylation by PP1.

Parallel results were obtained for the effect of subunit exchange on CaMKII switch robustness in the context of protein turnover. Turnover acts at a constant rate to replace any active CaMKII holoenzyme with an inactive holoenzyme (*Equation 6*), thus decreasing the stability of the ON state. Without subunit exchange, switch stability as measured by residence time of the ON state decreased exponentially with increasing turnover rate. With subunit exchange, however, residence time of the ON state remained roughly constant upto a ~10 fold increase in turnover (*Figure 3B*), after which subunit exchange could not phosphorylate all the inactive holoenzymes produced by turnover. At this point, the switch started to show a similar steep decay of stability as was seen without subunit exchange. As expected, turnover increased the number of switching events in the regime of bistability in both cases.

Thus, subunit exchange broadens the zone of bistability of the switch, both with respect to the range of $N_{PP1}$, and the turnover rate over which the switch remains bistable. It also reduces fluctuations in the ON state of the switch.

## Subunit exchange facilitates the spread of CaMKII activity

As suggested in *Stratton et al., 2014*, we found that subunit exchange facilitated the spread of CaMKII activation (*Figure 4*). When subunits were allowed to diffuse, an active subunit could be picked by a neighboring inactive CaMKII holoenzyme, making it partially phosphorylated. This process overcomes the first slow step of CaMKII phosphorylation (*Equation 1*), especially when subunit exchange makes many phosphorylated subunits available, thereby facilitating the spread of activation.

We simulated $N_{CaMKII}$ = 18 inactive holoenzymes in a cylindrical arena with a volume of 0.0275 µm³ and a length of 540 nm representing the PSD. The cylinder was divided into 18 voxels (one holoenzyme in each voxel). Each voxel was separated by 30 nm, which is the average nearest-neighbour distance for CaMKII holoenzymes (*Feng et al., 2011*). Each voxel was considered to be a well-mixed environment that is diffusion was instantaneous within the voxel. Between voxels, diffusion was implemented as cross-voxel jump reactions (see Materials and methods). We did not try 2D/3D diffusion because of its simulation complexity and because it would be expected to be qualitatively similar (*Fange et al., 2010*).

We fixed the diffusion coefficient of PP1 ($D_{PP1}$) to quantify the effect of varying the diffusion coefficient of subunits ($D_{sub}$) and basal calcium levels. We used $N_{PP1}$ = 0.5 µm² s⁻¹ which is the observed value of the diffusion coefficient of Ras, a similar sized protein (*Harvey et al., 2008*). We ran simulations for 4 h at basal calcium concentration [Ca²⁺] = 80 nM+$\epsilon$ (where $\epsilon$ is the fluctuation in basal calcium levels, see *Figure 1C*), and without subunit exchange (i.e. $D_{sub}$ = 0). We set $N_{PP1}$ = 15× $N_{CaMKII}$ to make sure the system showed no significant CaMKII activity (*Figure 4B*, red curve). This served as the baseline to quantify the effect of subunit exchange. When we enabled subunit exchange by

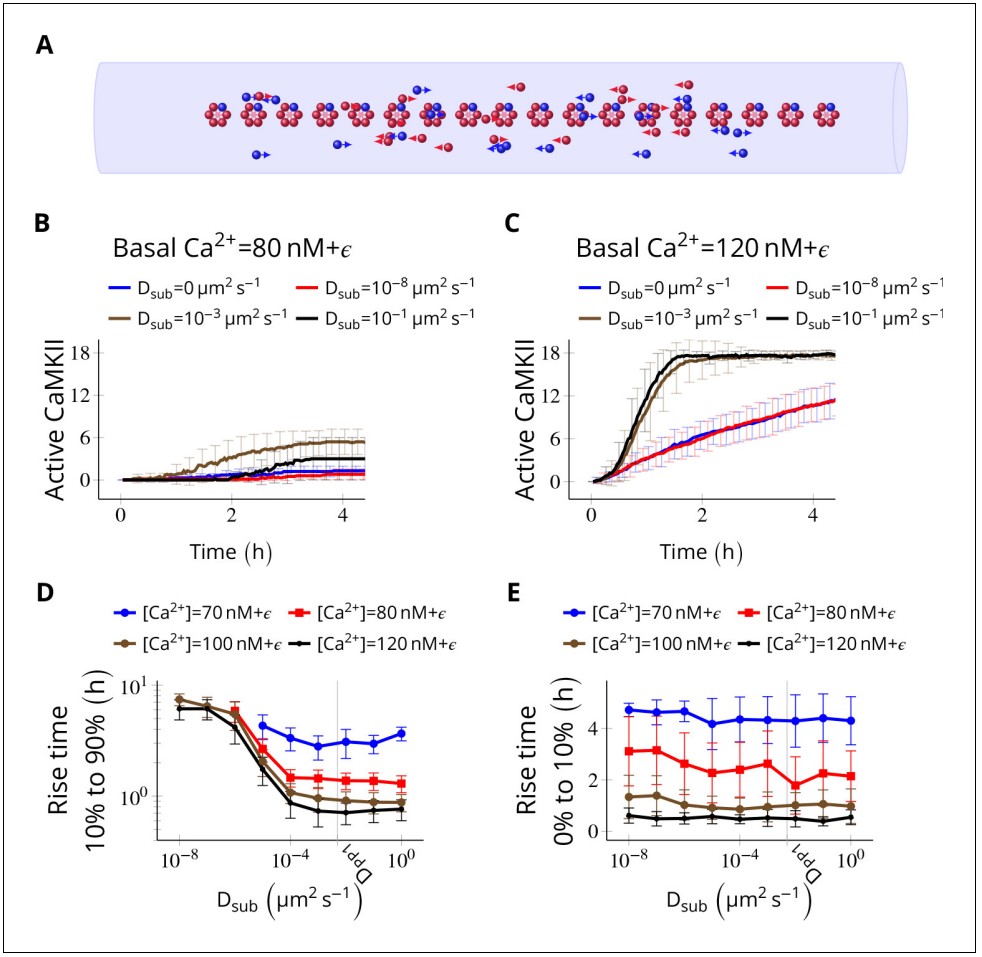

**Figure 4.** Subunit exchange facilitates the spread of kinase activity (*Stratton et al., 2014*). (**A**) 18 CaMKII holoenzymes were simulated in a cylindrical arena of volume 0.0275 $\mu m^3$, discretized into 18 voxels, each separated by 30 nm. Red and blue balls represent unphosphorylated and phosphorylated subunits, respectively. (**B**) Activation profile of CaMKII at mean basal calcium level of 80 nM+$\epsilon$ ($\epsilon$ is fluctuation in basal $Ca^{2+}$ levels *Figure 1A*) for various values of $D_{sub}$ with $N_{PP1} = 15\times N_{CaMKII}$. For this value of $N_{PP1}$, we see moderate or no mean activity of CaMKII for various values of $D_{sub}$ for basal $Ca^{2+} = 80$ nM + $\epsilon$. This serves as the baseline for comparisons. (**C**) At a slightly higher level of basal $Ca^{2+}$ (120 nM+$\epsilon$), subunit exchange has a stronger effect on CaMKII activation. When subunits were modeled with zero or very small diffusion coefficients ($D_{sub} = 0$ and $D_{sub} = 10^{-8}$ $\mu m^2$ $s^{-1}$), the effect of subunit exchange was smaller than when subunits were tested with moderate-to-high diffusion coefficients ($D_{sub} = 0.001$ and $0.1$ $\mu m^2$ $s^{-1}$), (**D**) Quantification of the effect of subunit exchange (shown in B and C) as measured by the time taken by CaMKII to rise from 10% to 90% of its maximum value (rise time) in hours vs $D_{sub}$ and basal $Ca^{2+}$ levels. The effect of subunit exchange is greater (i.e. shorter rise times) at higher calcium levels for all values of $D_{sub}$. Rise time is also shorter for larger $D_{sub}$ for all values of $[Ca^{2+}]$. Error bars represents standard deviation (n = 40 trajectories). (**E**) The time to onset of CaMKII activity is independent of $D_{sub}$ and depends only on $[Ca^{2+}]$. The time to onset of activity is measured as the time taken by inactive CaMKII to rise from 0 to 10% of its maximum value. Average time for the onset of activity decreased with increasing basal $[Ca^{2+}]$ levels but remained independent of $D_{sub}$ suggesting that subunit exchange does not play a significant role at the beginning of activation of CaMKII by $Ca^{2+}$. Error bar represents standard deviation (n = 40 trajectories). $D_{PP1} = 0.5$ $\mu m^2$ $s^{-1}$ for all simulations. Source data are available at https://github.com/dilawar/SinghAndBhalla_CaMKII_SubunitExchange_2018/tree/master/PaperFigures/elifeFigure4 (*Singh and Bhalla, 2018*).

DOI: https://doi.org/10.7554/eLife.41412.006

The following figure supplement is available for figure 4:

**Figure supplement 1.** Sample trajectories of CaMKII activation for basal $Ca^{2+}$ concentration of $100$ nM+$\epsilon$.

DOI: https://doi.org/10.7554/eLife.41412.007

setting $D_{sub}$ to a non-zero value, CaMKII activity rose to a maximum within 4 h even for a low value of $D_{sub} = 0.001$ µm² s⁻¹ (*Figure 4C*, black curve).

The first step of CaMKII phosphorylation (*Equation 1*) is slow since it requires binding of two calcium/calmodulin complex ($Ca^{2+}$/CaM) simultaneously (at basal $[Ca^{2+}] = 80$ nM$+\epsilon$ , $v_1 = 1.27 \times 10^{-5}$ s⁻¹). However, subunit exchange can also phosphorylate a subunit in a holoenzyme by adding an available phosphorylated subunit to it (*Equation 3*). Note that a $D_{sub}$ value as low as 0.001 µm² s⁻¹ is good enough for subunit exchange to be effective. With this value of $D_{sub}$, it takes roughly 0.9 s for the subunit to reach another holoenzyme which is, on average, 30 nm away. Under these conditions, the rate of picking up available active subunits (given in *Equation 3*) is faster than $v_1$. Expectedly, for larger $D_{sub}$ values (e.g., 0.001 and 0.1 µm² s⁻¹), subunit exchange becomes more effective (compare red and blue with the rest in *Figure 4D*).

As expected, at higher basal $Ca^{2+}$ levels (120 nM), the system showed higher CaMKII activity for all values of $D_{sub}$ (*Figure 4D*). Increasing $D_{sub}$ increased the effect of subunit exchange, as measured

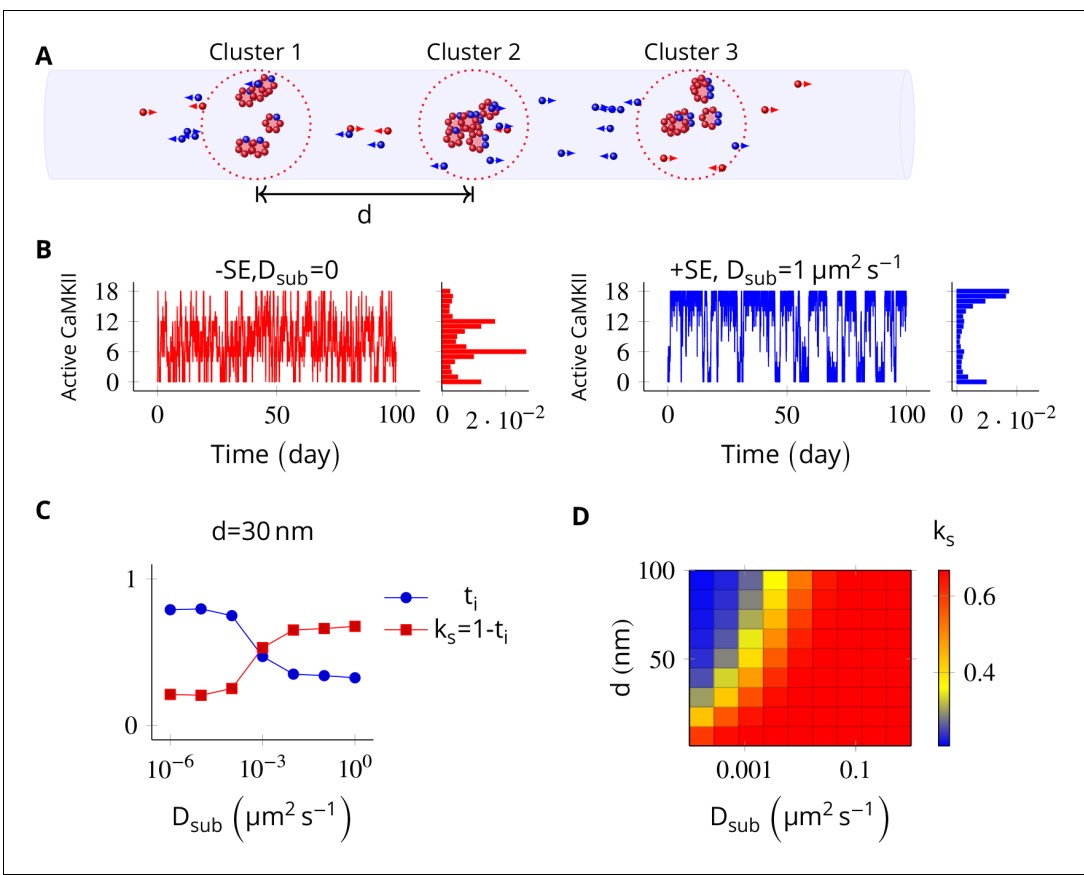

**Figure 5.** In the PSD, subunit exchange synchronizes activity of CaMKII clusters. (**A**) Three clusters, each of size 6 (i.e., $N_{CaMKII} = 6$) separated by distance $d$ were simulated in a cylindrical arena of volume 0.0275 µm³ discretized into three voxels. CaMKII subunits are shown as red (unphosphorylated) and blue (phosphorylated) balls. (**B**) (left) Without subunit exchange, all three switches flipped independently with a low residence time, resulting in a binomial distribution of states (Bar chart on right, in red). (right) With subunit exchange, all switches synchronized their activity that is they acted as a single bistable switch with a longer residence time. (**C**) Strength of synchronization ($k_s$) vs. diffusion constant $D_{sub}$ for a system consisting of three switches each separated from each other by a distance of 30 nm. Variable $k_s = 1 - t_i$ where $t_i$ is the fraction of total time spent by the switches in the intermediate states $x_a y_{n-a}$; $1 < a < n$. Synchronization is strong if $k_s > 0.4$. (**D**) 2-D plot of $k_s$ vs. $D_{sub}$ and $d$. The effect of synchronization $k_s$ due to subunit exchange is strong (red region) and robust to changes in $D_{sub}$, and effective for inter-cluster distance ($d$) as large as 100 nm. $D_{PP1} = 0.5$ µm² s⁻¹ for all simulations. Source data are available at https://github.com/dilawar/SinghAndBhalla_CaMKII_SubunitExchange_2018/tree/master/PaperFigures/elifeFigure5 (*Singh and Bhalla, 2018*).

DOI: https://doi.org/10.7554/eLife.41412.008

by the decreased rise time of CaMKII activity from 10% to 90% (*Figure 4D*). However, the time of onset of CaMKII activation as measured by rise time from 0% to 10% was dependent only on basal $Ca^{2+}$ levels but not on $D_{sub}$ (*Figure 4E*).

Thus, subunit exchange facilitates the spread of kinase activity following CaMKII activation but does not affect the onset of CaMKII activation.

## Subunit exchange synchronizes switching activity of clustered CaMKII

Next, we probed the effect of subunit exchange between spatially separated CaMKII clusters. We considered $N_{CaMKII}$ holoenzymes organized into three clusters of size $N_{CaMKII}/3$, each separated by a distance d. This configuration corresponds to cases where receptors and CaMKII holoenzymes are clustered at the synapse.

When there is no subunit exchange across voxels ($D_{sub}$ = 0), these switches are expected to switch independently like multiple coins flipped together, resulting in a binomial distribution of activity. The clustered system had three relatively stable bistable systems (long residence time, *Figure 1E*). As expected, without subunit exchange, activity in this system had a binomial distribution (*Figure 5B*, red plot).

Then we allowed PP1 and CaMKII subunits to undergo linear diffusion. We set $D_{PP1} = 0.5$ µm$^2$ s$^{-1}$ as before and varied $D_{sub}$ to quantify effect of subunit exchange. Subunit exchange led to synchronization of switching activity. The population of clustered CaMKII acted as a single bistable switch (*Figure 5B*, blue plot). This effect was strong and robust to variation in $D_{sub}$. Even for a very small value of $D_{sub}$ = 0.01 µm$^2$ s$^{-1}$, we observed strong synchronization (*Figure 5D*). The synchronization disappeared completely for $D_{sub}$ less than 0.0001 µm$^2$ s$^{-1}$, and for $d$ greater than 100 nm (*Figure 5D*).

Thus, for most physiologically plausible values of diffusion coefficient $D_{sub}$, subunit exchange causes synchronization of switching activity of clustered CaMKII.

## Subunit exchange may account for the observed dual decay rate of CaMKII phosphorylation

Finally, we asked if subunit exchange might account for the complex time-course of CaMKII dynamics in spine as observed in recent experiments (*Chang et al., 2017*). We designed a simulation to replicate an experiment where CaMKII was inhibited by a genetically encoded photoactivable inhibitory peptide after activating CaMKII by glutamate uncaging (*Murakoshi et al., 2017*). In the spine, CaMKII is more accessible to phosphatases than in the PSD, where our previous calculations had been located. To model the increased availability of phosphatases, we increased the concentration of PP1 by an order of magnitude, and increased the volume of the compartment to match the volume of a typical spine head that is 0.02 µm$^3$ (*Bartol et al., 2015*). We found that CaMKII acted as a leaky integrator of the calcium activity with a typical exponential decay dynamics (*Figure 6A*). We then enabled the diffusion of CaMKII subunits ($D_{sub}$ = 1 µm$^2$ s$^{-1}$) and PP1 ($D_{PP1}$ = 0.5 µm$^2$ s$^{-1}$). These conditions decreased the rate of dephosphorylation roughly by a factor of 5 (41.65 s vs. 200.82 s) (*Figure 6B*).

We expected that subunit exchange should have a strong effect on the time-course of decay of activity of clustered CaMKII in spine cytosol (e.g. CaMKII bound to actin) because the proximity of holoenzymes would lead to rapid exchange. Thus, if there are populations of clustered as well as non-clustered CaMKII in the spine, we expected that they would exhibit long and short time-courses of activity decay, respectively. Therefore a mixed population of clustered and non-clustered CaMKII should decay with two time-constants. Our simulations supported this prediction.

In *Chang et al. (2017)*, the decay kinetics of CaMKII were obtained by curve fitting of experimental data. It was given by a double-exponential function: $F(t) = P_{fast}e^{-t/\tau_{fast}} + P_{slow}e^{-t/\tau_{slow}}$ where $P_{fast} = 0.74$, $P_{slow} = 0.26$, $\tau_{fast} = 6.4 \pm 0.7$ s, $\tau_{slow} = 92.6 \pm 50.7$ s (*Figure 6C*, magenta). We used their estimate of $P_{fast}$ and $P_{slow}$ to construct a mixed population of slow and fast decaying CaMKII (*Figure 6A*, black), and simulated the decay kinetics of CaMKII for this system. We fit the resulting decay curve with a double-exponential function (*Figure 6C*, dashed red). The time-constants obtained (8.4 s, 86.2 s) matched well with experimentally estimated time-constants of (6.4 s ± 0.7, 92.6 s ± 50.7).

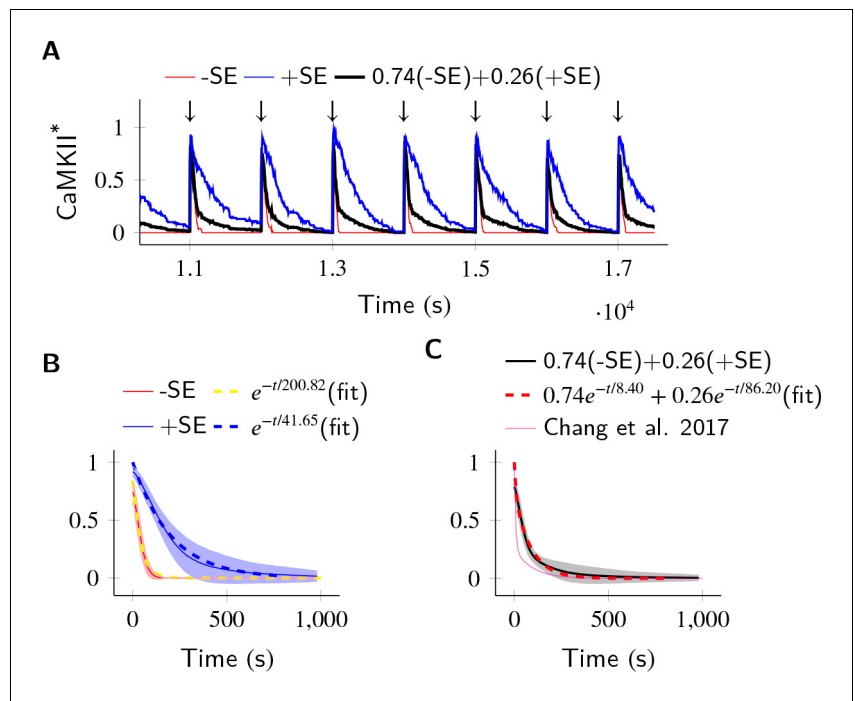

**Figure 6.** In the PP1-rich spine cytosol, CaMKII acts as a leaky integrator of $Ca^{2+}$ activity. The clustered CaMKII population decays more slowly than the non-clustered population, due to subunit exchange. Thus, a mixed population of clustered and non-clustered CaMKII can explain observed two time-constants of CaMKII decay (*Chang et al., 2017*). (A) Trajectories of CaMKII activity (fraction of all CaMKII which are active) when a strong periodic $Ca^{2+}$ pulse of 3 s duration was applied to the system after every 1000 s (↓). After the pulse, $Ca^{2+}$ levels were brought down to 80 nM. Three trajectories are shown: without subunit exchange (red), with subunit exchange (blue), and a weighed sum of red and blue (74% red +24% blue as estimated in [*Chang et al., 2017*]). (B) Average decay dynamics after the onset of strong $Ca^{2+}$ pulse (↓). When there was no subunit exchange, CaMKII decayed with a time-course of approximately 41.65 s (red and dashed yellow [fit]). When subunit exchange was enabled, CaMKII decay had a slower time-constant of 200.82 s (blue and dashed blue [fit]). (C) Average dynamics of the mixed population (black). This was fit to a double exponential that is $ae^{-t/\tau_1} + (1 - a)e^{-t/\tau_2}$ for $a = 0.74$ (dashed red). For a given $a = 0.74$ (estimated in [*Chang et al., 2017*]), our estimate of time-constants (8.4 s, 86.2 s) matched well with experimentally estimated time-constants (6.4 s ± 0.7, 92.6 s ± 50.7). Shaded areas are the standard deviation. Number of voxels Nv = 10, $D_{sub}$ = 1 $\mu m^2\ s^{-1}$, $D_{PP1}$ = 0.5 $\mu m^2\ s^{-1}$. Source data are available at https://github.com/dilawar/SinghAndBhalla_CaMKII_SubunitExchange_2018/tree/master/PaperFigures/elifeFigure6 (*Singh and Bhalla, 2018*).
DOI: https://doi.org/10.7554/eLife.41412.009

Thus, we suggest that subunit exchange may be a mechanism that leads to CaMKIIα activity decaying with two time-courses in spine cytosol.

## Discussion

Here, we have shown that subunit exchange strongly affects the properties of the CaMKII/PP1 pathway, both in its role as a bistable switch in the PSD and as a leaky integrator of $Ca^{2+}$ activity in spine cytosol. In the PSD, where the model was tuned to elicit bistable dynamics from clustered CaMKII, subunit exchange improved the stability of the CaMKII/PP1 switch by synchronizing the kinase activity across the PSD (*Figure 6*). It also improved active CaMKII tolerance of PP1, and of turnover rate (*Figure 2* and *Figure 3*). In the case where CaMKII was uniformly distributed in PSD, subunit exchange facilitated more rapid activation of CaMKII (*Figure 4B–D*) (*Stratton et al., 2014*). These simulation results predict that a CaMKII mutant lacking subunit exchange would be deficient in switch stability and slower to be activated by $Ca^{2+}$, thereby resulting in degraded memory retention and deficient learning in memory-related behavioral experiments, respectively.

**Table 1.** Diverse timescales of activity shown by CaMKII

| Type | Location | Timescale | Ref/Notes |
| --- | --- | --- | --- |
| Leaky integrator | Spine cytosol | ~10 s | *Chang et al., 2017*, This paper |
| Leaky integrator decaying slowly due to subunit exchange | Spine cytosol | ~100 s | This paper, *Chang et al., 2017* |
| Small bistable (Size 4 to 10) | PSD | Few hours to days | This paper |
| Large bistable (Size 12 to 20) | PSD | Few weeks to months | This paper |
| Synchronized population of bistables coupled by subunit exchange | PSD | Years | This paper |

DOI: https://doi.org/10.7554/eLife.41412.010

In the spine head, subunit exchange facilitated integration by prolonging the decay time-course of kinase activity (*Figure 6*). The fact that CaMKII dynamics changed from an integrator to bistable switch as we moved from spine cytosol (a phosphatase rich environment) to the PSD (where PP1 is tightly controlled) suggests an interesting sub-compartmentalization of functions in these microdomains. Furthermore, we observed that the clustering of CaMKII had important implications for its sustained activity.

Subunit exchange is unlikely to have any impact on neighbouring spines. The mean escape time of a single CaMKII subunit from a typical spine is between 8 s to 33 s (*Holcman and Schuss, 2011*). Any phosphorylated subunit is almost certain to be de-phosphorylated by PP1 during this time. We therefore predict that the effects of synchronization are local to each PSD, where PP1 is known to be tightly controlled. Subunit exchange loses its potency in the phosphatase rich region of the bulk spine head or dendrite. We therefore consider it unlikely that CaMKII subunit exchange plays any role in intra-spine information exchange such as synaptic tagging.

CaMKII is non-uniformly distributed in the PSD where it is mostly concentrated in a small region of 16 nm to 36 nm below the synaptic cleft (*Petersen et al., 2003*). In the PSD, CaMKII may exist in large clusters given that the PSD is rich in CaMKII binding partners. Our study predicts that subunit exchange may lead to synchronization when CaMKII is clustered, or more rapid activation by $Ca^{2+}$ when it is uniformly distributed. Given that CaMKII can form clusters with N-methyl-D-asparate (NMDA) receptors, it would be interesting to study the mixed case where some CaMKII is clustered and the rest is uniformly distributed. This would require a detailed 3D simulation and is beyond the scope of this study.

Finally, we suggest that the existence of diverse time-scales of CaMKII activity – bistable and highly stable synchronized bistable in PSD, slow and fast decaying leaky integrator in spine head (*Table 1*) – has important theoretical implications. A very plastic synapse is good at registering activity dependent changes (learning) but poor at retaining old memories. On the other hand, a rigid synapse is good at retaining old memories but is not efficient at learning. A theoretical meta-model which sought to strike a balance between these two competing demands requires that a diversity of timescales must exist at the synapse (*Benna and Fusi, 2016*) for optimum performance. In this model, complex synapses with state variables with diverse time-scales are shown to form a memory network in which storage capacity scales linearly with the number of synapses, and memory decay follows $1/\sqrt{t}$ — a power-law supported by psychological studies (*Wixted and Ebbesen, 1991*). This model requires the memory trace to be first stored in a fast variable and then progressively and efficiently transferred to slower variables. Our study suggests a concrete mechanism for such a process. Here, the $Ca^{2+}$ concentration in the PSD can be mapped to the fastest variable. The CaMKII integrator in the cytosol could represent the second slower variable to which the trace is transferred from $Ca^{2+}$. Further, the state information is transferred to the third slower CaMKII bistable switch. The dynamics of CaMKII in the PSD forms an even slower bistable variable for longer retention of the memory trace. It is possible that memory is transferred from here to even slower variables, such as sustained receptor insertion (*Hayer and Bhalla, 2005*), PKM-$\zeta$ activation (*Sacktor, 2012*), or local protein synthesis (*Aslam et al., 2009*).

## Materials and methods

We extended Miller and Zhabotinksy (MZ model; *Miller et al., 2005*) to incorporate *subunit exchange* and diffusion. We assume that vertical dimers are inserted and released together

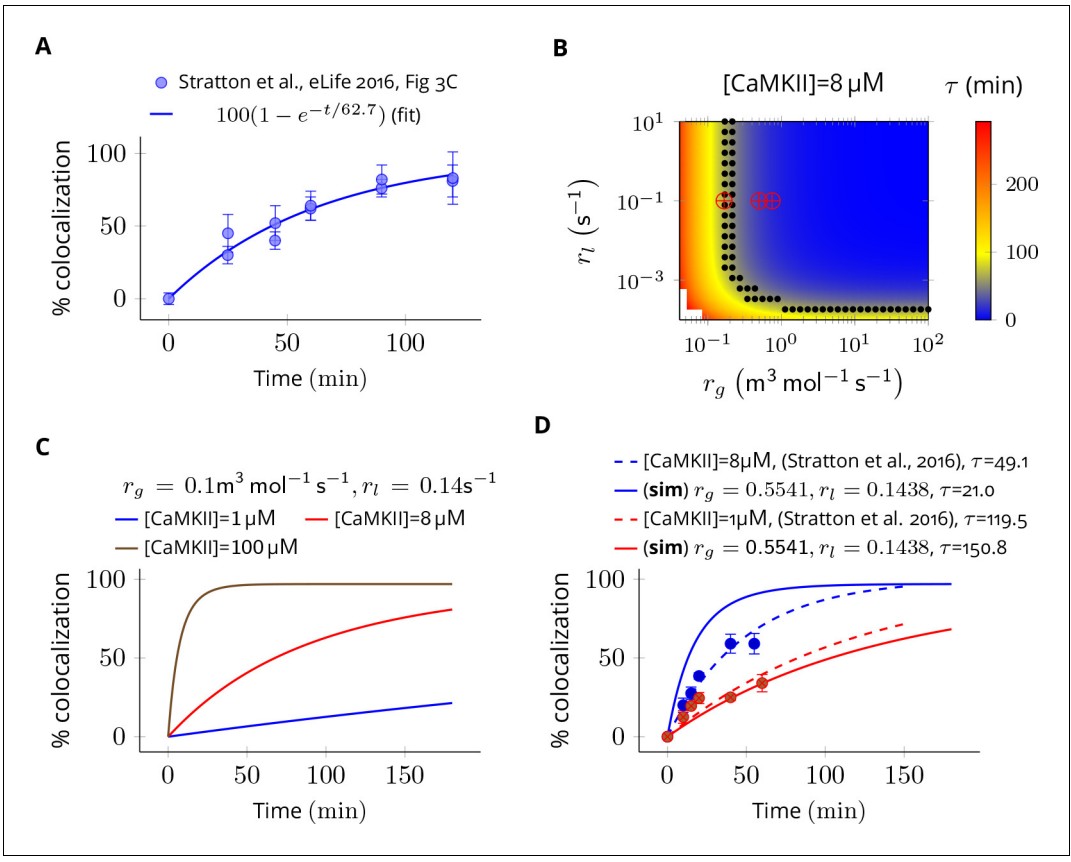

**Figure 7.** Estimation of subunit exchange rate by fitting experimental data to a model of a single molecule assay (*Stratton et al., 2014*). (**A**) Colocalization dynamics as reported in *Stratton et al., 2014* (all data scraped from figures) at CaMKII = 8 µM (blue dots). Solid blue line shows a best fit $100(1 - e^{-t/\tau})$ with τ=62.7 min. (**B**) Phase plot of τ of colocalization trajectories generated for various values of $r_g$ and $r_l$ (*Equation 4*). Black dots show values of $r_g$ and $r_l$ for which τ = 62.7 ± 20% (S.E.M.). Red ⊕ marks show the values of rate constants (*Equation 3*) used in this study at various volumes and $N_{CaMKII}$. (**C**) For the fixed values of $r_g$ and $r_l$, three trajectories are shown at different CaMKII concentrations. As seen in the experimental data, the rate of colocalization increases with increasing CaMKII concentration. (**D**) For typical values of exchange rates used in this paper, we plotted simulation results (solid lines) with experimental values (dots) and their best exponential fit (dashed lines). The $d\tau/d[\text{CaMKII}]$ was −10.06 min/µM (data) and −18.54 min/µM (simulation). Source data are available at https://github.com/dilawar/SinghAndBhalla_CaMKII_SubunitExchange_2018/tree/master/PaperFigures/elifeFigure7 (*Singh and Bhalla, 2018*).
DOI: https://doi.org/10.7554/eLife.41412.011

(*Bhattacharyya et al., 2016*). We also assume that both subunits of a vertical dimer phosphorylate and de-phosphorylate together. Under this assumption, we can treat the CaMKII ring as the proxy for the CaMKII holoenzyme and the subunit as the proxy for the CaMKII dimer. Without this assumption, the simulation cost of the increased complexity would be very significant.

In our model, a CaMKII ring with $n$ subunits ($n$ = 6 or 7) can exist in 15 different states enumerated as $x_a y_{n-a}$ for $0 \le a \le n$ where $x$ and $y$ represent un-phosphorylated and phosphorylated subunits respectively. We ignore all rotational permutations and kinetically unlikely cases where there are discontiguous phosphorylated subunits in the ring. We assumed that the phosphorylation of neighbouring subunit proceeds clockwise.

## Ca²⁺ background activity ($\epsilon$)

We assumed the resting Ca²⁺ level in spine to be 80 nM (*Berridge, 1998*). In the MZ model, Miller *et al.* assumed that Ca²⁺ entry through NMDA receptors can be approximated by a Poisson train with an average rate of 0.5 Hz. Since, on average, ∼0.5 NMDA receptors open (*Nimchinsky et al., 2004*) upon pre-synaptic stimulation, we reduced the frequency of NMDA opening events to

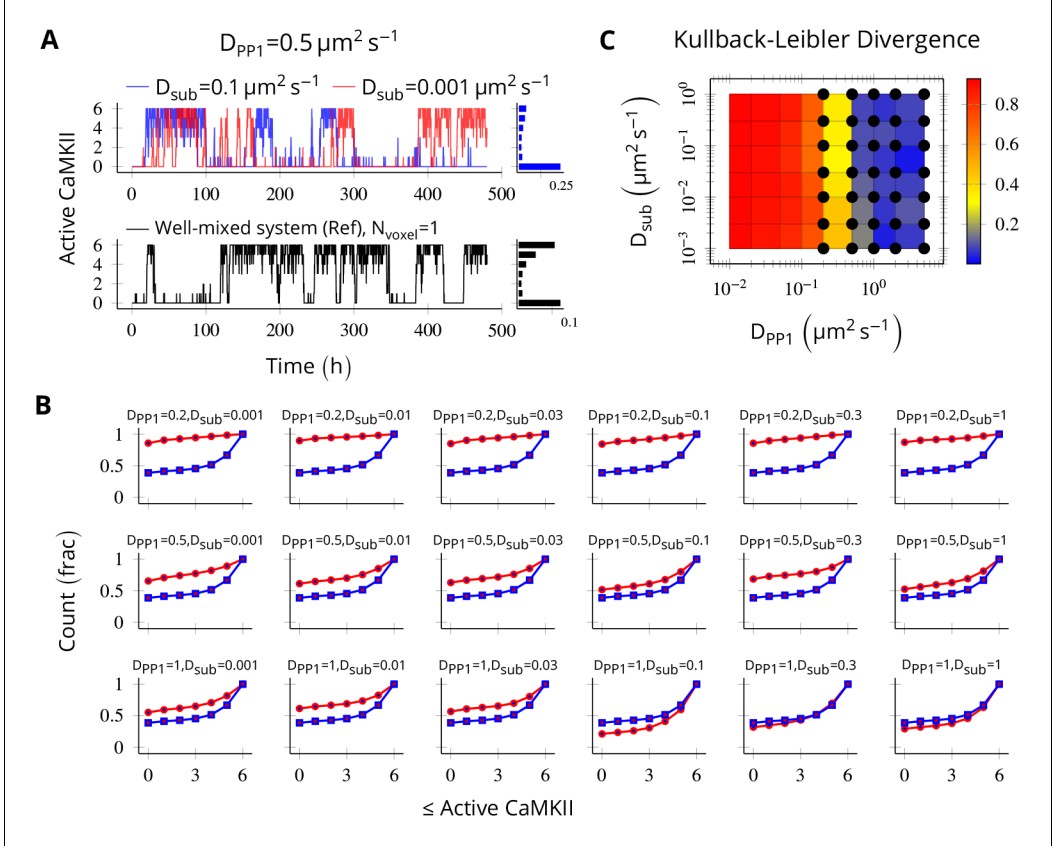

**Figure 8.** Method validation. NCaMKII = 6 holoenzymes as described in *Figure 4* were simulated in a cylindrical arena divided into six voxels separated by 30 nm. Basal $Ca^{2+}$ was set to 100 nM+ε. (**A**, above) For a typical value of $D_{PP1}$ = 0.5 $\mu m^2 s^{-1}$ used in our model, varying $D_{sub}$ did not result in loss of bistability of CaMKII activity. The distribution of state occupancy is shown for the case of $D_{sub}$ = 0.1 $\mu m^2 s^{-1}$ in bar chart on the right. (**A**, below) Reference well-mixed system for comparison. Six holoenzyme were simulated in a single well-mixed cylinder of same length and volume, and with same parameter values as above. The distribution of state occupancy is shown on the right. (**B**) Cumulative histograms of CaMKII activity for various values of $D_{PP1}$ and $D_{sub}$ (unit $\mu m^2 s^{-1}$). The red and blue lines represent spatially discretized and well-mixed reference system (shown in A, black) respectively. The spatially discretized system (red) converges to the well-mixed system (blue) for higher $D_{PP1}$ values. For fixed value of $D_{PP1}$, changing $D_{sub}$ has little or no effect on convergence. For a typical value of $D_{PP1}$ = 0.5 $\mu m^2 s^{-1}$, the system shows reasonable convergence (second row, also see **A**). (**C**) Quantification of convergence to well-mixed case. We used Kullback-Leibler divergence (relative entropy) to quantify the similarity between the state occupancy histograms (e.g. as in panel **A**) for the spatially extended case, and the reference well-mixed system, respectively. Identical histograms will have zero Kullback-Leibler divergence. The phase plot shows Kullback-Leibler divergence between the histograms for the spatially extended system and the reference bistable system. Black dots represent bistable configurations with at least four transitions observed in a simulation of the spatially discretized system, lasting 20 days. Thus, the spatially extended, discretized system converged to the behavior of the reference bistable system. Source data are available at https://github.com/dilawar/SinghAndBhalla_CaMKII_SubunitExchange_2018/tree/master/PaperFigures/elifeFigure8 (*Singh and Bhalla, 2018*).

DOI: https://doi.org/10.7554/eLife.41412.012

The following figure supplement is available for figure 8:

**Figure supplement 1.** PP1 potency reduces as $D_{PP1}$ increases.

DOI: https://doi.org/10.7554/eLife.41412.013

0.25 Hz. We used a periodic pulse with a time-period of 4 s and duty cycle of 50%. To model NMDA activity in the 2 s long ON period of our 4 s long periodic pulse, we sampled from a uniform distribution with median of 120 nM (50% change, on average) and range of 40 nM (*Figure 1C*). This distribution is informed by *Figure 2B,C* from (*Nimchinsky et al., 2004*).

We did not consider decay dynamics of $Ca^{2+}$ influx through the NMDA channel since the time-scale of decay (roughly 100 ms) is much shorter than our simulation runtimes of days, and including this detail would have made the simulations very slow. The effect of ignoring decay dynamics are expected to be negligible given that the time-scale of CaMKII activation is much larger than the

time course of $Ca^{2+}$ decay dynamics. Furthermore, we did not consider contributions to background $Ca^{2+}$ fluctuations by other channels. This background activity is represented by $\epsilon$ in the figures and text.

## Phosphorylation and dephosphorylation of CaMKII ring

The activation of CaMKII in our study follows the same dynamics as in the MZ model (*Equation 1*). In our paper, by phosphorylation/activation of a CaMKII subunit or a holoenzyme, we mean phosphorylation at Thr[286]. The first step in CaMKII activation requires simultaneous binding of two ($Ca^{2+}$/CaM) to the two adjacent subunits of CaMKII. Once a subunit is phosphorylated, it catalyzes phosphorylation of its neighbour (*auto-phosphorylation*) which requires binding of only one ($Ca^{2+}$/CaM). Therefore, further phosphorylation proceeds at much faster rate. The phosphorylation of CaMKII is given by *Equation 1* (*Bradshaw et al., 2003*; *Miller et al., 2005*).

$$x_a y_{n-a} \xrightarrow{v_1} x_{a-1} y_{n-a+1} \xrightarrow{v_2} x_{a-2} y_{n-a+2}$$
$$v_1 = k_1 \left[\frac{H^3}{1+H^3}\right]^2, v_2 = k_1 \frac{H^3}{1+H^3}, \text{ where } H = \frac{Ca^{2+}}{K_{H1}}$$

(1)

where $n = 6$ or $7$ for $1 \le a \le n$; $k_1 = 1.5\,\text{s}^{-1}$ (*Hanson et al., 1994*), and $k_{H1} = 0.7\,\mu M$ (*De Koninck and Schulman, 1998*; *Miller et al., 2005*). At resting $Ca^{2+}$ concentration of 100 nM, $v_1 = 1.27 \times 10^{-5}\,\text{s}^{-1}$ and $v_2 = 4.36 \times 10^{-3}\,\text{s}^{-1}$ (i.e., $v_2/v_1 \approx 343$). The rate constant $v_1$ above defines the initial rate of phosphorylation. Furthermore, addition of phosphorylated subunits can happen through subunit exchange (*Equation 3*). We treat these as independent variables. The phosphorylation rates $v_1$ and $v_2$ are relatively well constrained by the experimental literature. The subunit exchange rates were estimated (Materials and methods) to be in the range of 1 s⁻¹.

Once fully phosphorylated, CaMKII moves to the PSD where it binds to the NMDA receptor. Upon binding, it is no longer accessible other phosphatases except PP1.

The dephosphorylation of the CaMKII ring, and the subunit are given by *Equation 2*.

$$PP1 + x_a y_{n-a} \underset{k^-}{\overset{k^+}{\rightleftharpoons}} PP1.x_a y_{n-a} \xrightarrow{k_2} PP1 + x_{a+1} y_{n-a-1}$$
$$PP1 + x \underset{k^-}{\overset{k^+}{\rightleftharpoons}} PP1.x \xrightarrow{k_2} PP1 + y$$

(2)

where $n = 6$ or $7$, and $1 \le a \le n$. Following (*Miller et al., 2005*), we also assumed $k^- = 0$. This gave us $k^+ = \frac{k_2}{k_M} = 1/\mu M/s$. We could not find any experimental estimate of $k_M$ in recent literature, therefore we used the same value of $k_M$ as in the MZ model (*Miller et al., 2005*).

## Subunit exchange

Since CaMKII ring consists of either 6 or 7 subunits in our model, any ring with six subunits cannot lose a subunit, and a ring with seven subunits cannot gain a subunit. The reactions which result in either gain or loss of a subunit are given by *Equation 3* where $0 \le a \le 6\,\text{or}\,7$.

$$x_a y_{7-a} + x \underset{k_x^-}{\overset{k_x^+}{\rightleftharpoons}} x_{a+1} y_{6-a}$$
$$x_a y_{6-a} + y \underset{k_y^-}{\overset{k_y^+}{\rightleftharpoons}} x_a y_{7-a}$$

(3)

We were not able to find values for $k_x^+$, $k_x^-$, $k_y^+$, and $k_y^-$ in the literature. We used the data in *Stratton et al. (2014)* to estimate the possible timescale of subunit exchange rate. (*Bhattacharyya et al., 2016*) speculate that upon activation, the hub of the holoenzyme becomes less stable and more likely to open up and lose a subunit that is an active holoenzyme loses subunits at a faster rate. Therefore, we maintained the following ratio $k_x^- \approx 10 k_x^+ N_{\text{CaMKII}}$ and $k_y^- \approx 10 k_y^+ N_{\text{CaMKII}}$ in all simulations where $N_{\text{CaMKII}}$ is the number of holoenzymes in the system.

**Table 2.** Table of parameters used in model.

| Symbol | Parameter | Value | Reference/Notes |
|---|---|---|---|
| $V_{spine}$ | Volume of Spine | 1 to $5\times10^{-20}$ m$^3$ | (*Bartol et al., 2015*) |
| $V_{PSD}$ | Volume of PSD (Thickness 100 nm. Surface area 0.05µm2) | 1 to $5\times10^{-21}$ m$^3$ | *Farley, 2015*; *Bartol et al., 2015* |
| $N_{CaMKII}$ | Total CaMKII holoenzymes in PSD/Spine | 100 ± 18 | *Farley, 2015* |
| $N_{PP1}$ | Total PP1 in PSD <br> Total PP1 in Spine | 4 to 20× $N_{CaMKII}$ <br> 10 to 100× $N_{CaMKII}$ | This paper <br> This paper |
| I1 | Concentration of free I1 | 10 µM | *Miller et al., 2005* |
| $V_{CaN}$ | Activity of calcineurin divided by its | 1 | *Miller et al., 2005* |
| $V_{CaM}$ | Activity of PKA divided by its Michaelis constant | 1 s$^{-1}$ | *Miller et al., 2005* |
| KM | The Michaelis constant of PP1 | 10 µM | 0.4 to 20 µM (*Zhabotinsky, 2000*) |
| $K_{H1}$ | Hill constant of CaMKII (Ca$^{2+}$) activation | 0.7 µM | (*De Koninck and Schulman, 1998*) |
| $n_{H1}$ | Hill constant of CaMKII (Ca$^{2+}$) activation | 3 | (*Stemmer and Klee, 1994*) |
| $K_{H2}$ | Hill constant of CaMKII (Ca$^{2+}$) activation | 0.3 µM | (*Stemmer and Klee, 1994*) |
| $n_{H2}$ | Hill constant of CaMKII (Ca$^{2+}$) activation | 3 | (*Stemmer and Klee, 1994*) |
| k1 | The catalytic constant of autophospho-rylation | 1.5 s$^{-1}$ | (*Hanson et al., 1994*) |
| k2 | The catalytic constant of autophospho-rylation | 1 s$^{-1}$ | (*Bradshaw et al., 2003*; *Ichikawa et al., 1996*) |
| k3 | The association rate constant of PP1.I1P complex | 100 µM$^{-1}$s$^{-1}$ | (*Endo et al., 1996*; *Miller et al., 2005*) |
| k4 | The dissociation rate constant of PP1.I1P complex | 0.1 s$^{-1}$ | (*Endo et al., 1996*; *Miller et al., 2005*) |
| $k_x^+$ | The rate of adding unphosphorylated subunit x | 1 s$^{-1}$ per $N_{CaMKII}$ | This paper |
| $k_y^+$ | The rate of adding phosphorylated sub- unit y | 1 s$^{-1}$ per $N_{CaMKII}$ | This paper |
| $k_x^-$ | The rate of losing unphosphorylated subunit x | 0.1 s$^{-1}$ | This paper |
| $k_y^-$ | The rate of losing phosphorylated sub- unit y | 0.1 s$^{-1}$ | This paper |
| vt | Turnover rate of CaMKII | 30 h$^{-1}$ | (*Ehlers, 2003*; *Miller et al., 2005*) |
| $D_{PP1}$ | Diffusion coefficient of PP1 | 0.5 µm$^2$ s$^{-1}$ | This paper and (*Harvey et al., 2008*) |
| $D_{sub}$ | Diffusion coefficient of CaMKII subunits | $10^{-5} - 10$µm$^2$ s$^{-1}$ | This paper |

DOI: https://doi.org/10.7554/eLife.41412.014

## Estimation of subunit exchange rate

To estimate reaction rates of *Equation 3*, we modeled the 'single molecule assay' used in *Stratton et al., 2014*. In this assay, two distinct populations of CaMKII labelled by either green or red fluorophores were mixed together. The holoenzymes were not free to move but they could release subunits which could move freely. A green holoenzyme may pick up a red subunit and vice versa thereby giving rise to a mixed colored population. The readout from this assay is the 'colocalization' which is the fraction of total holoenzymes containing subunits of both colors.

In our model of this assay, a CaMKII holoenzyme is represented by $R_aG_{n-a}$ where $R$ and $G$ represent a red and a green subunit in the holoenzyme respectively, and n = 6 or 7. The green population consists of holoenzymes with only green subunits (i.e., $R_0G_6$ and $R_0G_7$) and the red population has holoenzymes with all red subunits (i.e., $R_6G_0$ or $R_7G_0$). We assume that each color population has equal number of dodecameric (n = 6 × 2) and tetradecameric (n = 7 × 2) holoenzymes. Upon mixing red and green populations, the following reactions take place.

$$R_aG_b \underset{r_g}{\overset{r_l}{\rightleftharpoons}} R_{a-1}G_b + R \quad \text{for all } a > 0, b \geq 0 \text{ s.t. } a+b = 6$$

$$R_aG_b \underset{r_g}{\overset{r_l}{\rightleftharpoons}} R_aG_{b-1} + G \quad \text{for all } a \geq 0, b > 0 \text{ s.t. } a+b = 7$$

(4)

The value of colocalization is equal to the percentage of all holoenzymes containing at least one

red and one green subunit that is $\frac{\sum_{a\geq 1, b\geq 1}[R_a G_b]}{\sum_{a\geq 0, b\geq 0}[R_a G_b]}$. The dynamics of colocalization was fit by $100(1 - e^{-t/\tau})$. We first computed $\tau$ for experimental data when [CaMKII] = 8 μM (*Figure 7A*). This served as the baseline for further analysis.

Next, we explored the space of $r_l$ and $r_g$ for which the time constant $\tau$ of colocalization dynamics matched well with the baseline case (i.e. $\tau$ for these trajectories were $\tau_{[CaMKII]}$ = 8 ± 20% (*Figure 7B*, black dots). From these values, we chose a combination of $r_g$ and $r_l$ which best explained the concentration-dependent changes in the rate of colocalization (*Figure 7D*). When compared with the data from *Stratton et al., 2014*), the time scale of colocalization and the concentration-dependent decrease in the rate of colocalization matched reasonably well for $r_l$ and $r_g$ that is, $\tau$ = 49.1 min (data) vs $\tau$ = 21.0 min (simulation) when [CaMKII] = 8 μM and $\tau$ = 119.0 min (data) vs $\tau$ = 150.0 min (simulation) when [CaMKII] = 1 μM, and, $\frac{d\tau}{d[CaMKII]}$ = -10.06 min/μM (data) vs $\frac{d\tau}{d[CaMKII]}$ = -18.05 min/μM (simulation) (*Figure 7D*). Note that we do not model the effect of diffusion, labelling efficiency, and experimental errors in the readout mechanism. Our values of $k_x^+, k_y^+, k_x^-, k_y^-$ used in *Equation 3* are close to estimated values of $r_g$ and $r_l$ (red cross vs. black dots in *Figure 7B*). Note that $r_g$ and $r_l$ are proxies for $k_x^+, k_y^+$ and $k_x^-, k_y^-$ respectively.

Thus, we are confident that rate parameters used in *Equation 3* in our model are likely to be within the physiologically relevant range.

## PP1 deactivation

In the PSD, PP1 is the primary – and perhaps only – phosphatase known to dephosphorylate CaMKII (*Strack et al., 1997b*). We followed the MZ model for *Equation 5* where inhibitor-1 (I1) inactivates PP1. Phosphorylated inhibitor-1 (I1P) renders PP1 inactive by forming I1P-PP1 complex (I1P.PP1).

$$PP1 + I1P \underset{k_4}{\overset{k_3}{\rightleftharpoons}} I1P.PP1$$

$$I1P = I1 \frac{v_{PKA}}{v_{CaN}} \frac{1 + \left(\frac{Ca}{k_{H2}}\right)^3}{\left(\frac{Ca}{k_{H2}}\right)^3} \tag{5}$$

where $k_3$ = 100 /μM/s, $k_4$ = 0.1 s$^{-1}$ (*Endo et al., 1996*), and $v_{PKA}/v_{CaN}$ = 1 (*Miller et al., 2005*).

## Turnover

The turnover of CaMKII is a continuous process given by *Equation 6* with rate $v_t = 30 h^{-1}$ (*Ehlers, 2003*).

$$x_a y_{6-a} \xrightarrow{v_t} x_6 y_0 \text{ for } 6 \geq a \geq 1$$
$$x_a y_{7-a} \xrightarrow{v_t} x_7 y_0 \text{ for } 7 \geq a \geq 1 \tag{6}$$

## Diffusion and simulation method

Diffusion is implemented as a cross voxel jump reaction. Diffusion of a species X with diffusion-coefficient $D_X$ between voxel A and B separated by distance $h$ is modelled as a reaction $X_A \underset{k}{\overset{k}{\rightleftharpoons}} X_B$ where $k = D_X/h^2$, and $[X_A] = [X_B] = [X]/2$ (*Erban et al., 2007*). Based on our own numerical results (*Appendix 1—figure 2*) and other studies (*Isaacson, 2009*; *Erban and Chapman, 2009*), we are confident that $h \geq 10 h_{crit}$ where $h_{crit} = \frac{k^+}{D_{PP1} + D_{sub}}$ is a good value. We have $h_{crit} \leq 3.2$ nm whenever $D_{PP1} + D_{sub} \geq 0.5$ μm$^2$ s$^{-1}$. For all simulations presented in main text, we maintained $h \geq h_{crit}$. For a few illustrative examples where $h$ is smaller than $h_{crit}$, see *Figure 4—figure supplement 1D,E*.

All simulations were performed using the stochastic solver based on the Gillespie method, in the MOOSE simulator (https://moose.ncbs.res.in, version 3.1.4; *Ray and Bhalla, 2008*). This model is available at https://github.com/dilawar/SinghAndBhalla_CaMKII_SubunitExchange_2018 (copy archived at https://github.com/elifesciences-publications/SinghAndBhalla_CaMKII_SubunitExchange_2018). The table of parameters is in SI (*Table 2*).

## Method validation

To validate our implementation of diffusion, we compared trajectories of two systems: one in a single well-mixed cylinder with parameters tuned to elicit bistable behavior (henceforth, we call it the reference bistable), and a spatial system implemented as a discretized cylinder as described above. We expect the later to converge to reference bistable system when the diffusion constants become large such that the molecules are effectively well-mixed.

We put six CaMKII holoenzymes in a cylinder of length 180 nm discretized into six voxels, separated by a distance of 30 nm. The long-term behavior of discretized system was most sensitive to $D_{PP1}$ (*Figure 8B*) and almost independent of $D_{sub}$ (*Figure 8A*). The discretized system converges to reference bistable for $D_{PP1} \geq 0.5\mu m^2 s^{-1}$ (*Figure 8C*).

## Table of parameters

*Table 2* summarizes the parameters of our model.

## Acknowledgements

We thank Marcus Benna, Stefano Fusi and Moitrayee Bhattacharyya for discussions related to their work, Mukund Thattai for useful discussions on the stochastic reaction diffusion methods, and Bhanu Priya for useful comments on the manuscript. This work was funded by NCBS/TIFR and SERB JC Bose fellowship SB/S2/JCB-023/2016 to USB.

## Additional information

### Competing interests

Upinder Singh Bhalla: Reviewing editor, *eLife*. The other author declares that no competing interests exist.

### Funding

| Funder | Grant reference number | Author |
| --- | --- | --- |
| Science and Engineering Research Board | JC Bose Fellowship #SB/S2/JCB-023/2016 | Upinder Singh Bhalla |

The funders had no role in study design, data collection and interpretation, or the decision to submit the work for publication.

### Author contributions

Dilawar Singh, Conceptualization, Resources, Software, Formal analysis, Investigation, Visualization, Writing—original draft, Writing—review and editing; Upinder Singh Bhalla, Conceptualization, Resources, Software, Supervision, Funding acquisition, Validation, Methodology, Project administration, Writing—review and editing

### Author ORCIDs

Dilawar Singh http://orcid.org/0000-0002-4645-3211
Upinder Singh Bhalla http://orcid.org/0000-0003-1722-5188

### Decision letter and Author response

Decision letter https://doi.org/10.7554/eLife.41412.020
Author response https://doi.org/10.7554/eLife.41412.021

## Additional files

### Supplementary files

• Transparent reporting form
DOI: https://doi.org/10.7554/eLife.41412.015

### Data availability

The model and the instructions to generate data analysed in this study are available at https://github.com/dilawar/SinghAndBhalla_CaMKII_SubunitExchange_2018 (copy archived at https://github.com/elifesciences-publications/SinghAndBhalla_CaMKII_SubunitExchange_2018).

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

## Appendix 1

DOI: https://doi.org/10.7554/eLife.41412.016

### Stochastic diffusion using cross-voxel reactions

To implement diffusion of a molecule in a linear arena of volume $V$, we constructed a cylindrical compartment of volumne $V$ with length $L$ and radius $r$. The cylinder was divided into $n$ well-mixed voxels of length $h$ ($h = L/n$) i.e., within a voxel, diffusion was instantaneous.

Diffusion across these voxels was implemented as cross-voxel reactions in which a molecule jumps to its neighbouring voxels with a rate constant $k$ which is a function of diffusion coefficient $D$ and length of voxel $h$. The accuracy of this method increases as $h$ decreases and converges to the analytical solution for $\lim_{h \to 0}$. The cost of simulation increases as $h$ decreases.

For example, assume that molecule A with diffusion coefficient of $D_A$ is put into a cylinder. We divide the entire cylindrical volume into $n$ voxels. We uniformly distribute all molecules of A into these $n$ voxels. Any molecule from voxel $i$ (labelled $A_i$) can only jump to the neighbouring voxel $i+1$ or $i-1$ with rate $k_D^A$. This process is described by the following chemical reactions (*Equation 7*).

$$\ldots A_{i-1} \underset{k_D^A}{\overset{k_D^A}{\rightleftharpoons}} A_i \underset{k_D^A}{\overset{k_D^A}{\rightleftharpoons}} A_{i+1} \ldots \tag{7}$$

### Stochastic diffusion with bimolecular reaction

Consider a cylindrical arena with a simple bimolecular reaction $A + B \to \phi$. Both A and B diffuse while $\phi$ does not. In our model, this reaction resembles dephosphorylation of subunit or CaMKII holoenzyme by PP1 . If we divide the cylinder into $n$ voxels labelled 1, 2, ..., $n$, then we have the following resultant chemical system.

$$A_1 + B_1 \overset{k}{\to} \phi_1$$
$$A_2 + B_2 \overset{k}{\to} \phi_2$$
$$\vdots$$
$$A_n + B_n \overset{k}{\to} \phi_n$$

$$A_1 \underset{k_D^A}{\overset{k_D^A}{\rightleftharpoons}} A_2, A_2 \underset{k_D^A}{\overset{k_D^A}{\rightleftharpoons}} A_3, \ldots, A_{n-1} \underset{k_D^A}{\overset{k_D^A}{\rightleftharpoons}} A_n$$

$$B_1 \underset{k_D^B}{\overset{k_D^B}{\rightleftharpoons}} B_2, B_2 \underset{k_D^B}{\overset{k_D^B}{\rightleftharpoons}} B_3, \ldots, B_{n-1} \underset{k_D^B}{\overset{k_D^B}{\rightleftharpoons}} B_n$$

$$(8)$$

where $k_D^A = \frac{D_A}{h^2}$ and $k_D^B = \frac{D_B}{h^2}$ and $[A_1] = [A_2] = \ldots = [A_n] = [A]$, $[B_1] = [B_2] = \ldots = [B_n] = [B]$ and where $[X]$ is the concentration of X.

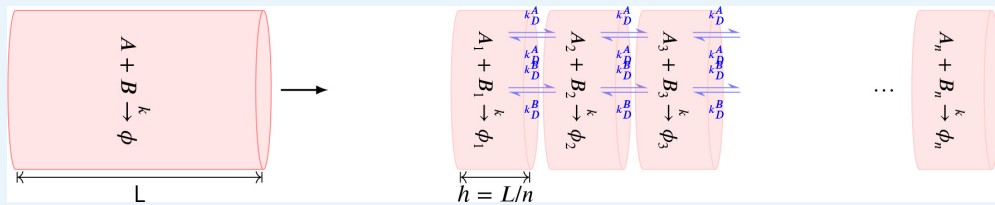

**Appendix 1—figure 1.** Summary of the stochastic reaction diffusion method employed.

DOI: https://doi.org/10.7554/eLife.41412.017

For a bimolecular system described by *Equation 8* when solved using the Gillespie algorithm, as $h$ decreases, the accuracy of the solution first increases and subsequently starts

decreasing. This is because with decreasing $h$, the diffusion component becomes more precise but we start losing reaction events (*Gardiner et al., 1976*). Analytical methods suggests a lower bound on $h$, namely $h \gg \frac{k}{D_A+D_B}$ below which errors in the solution are large (*Isaacson, 2009*). A numerical study (*Erban and Chapman, 2009*) suggests that $h \geq 10\frac{k}{D_A+D_B}$ is a good value which keeps errors less than 1% in the distributions of trajectory.

Here, we simulate the system and estimate relative errors for various values of diffusion coefficient (*Appendix 1—figure 2*).

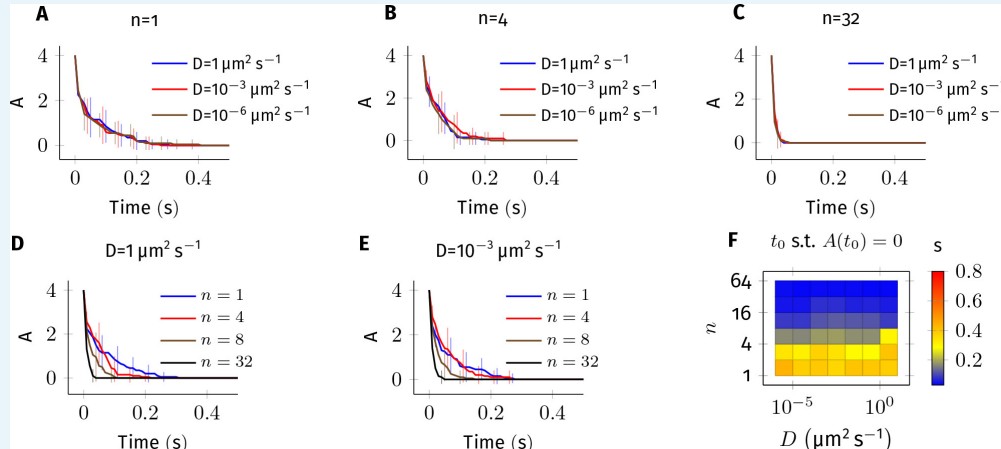

**Appendix 1—figure 2.** Error estimate for the bimolecular reaction $A + B \xrightarrow{k} \phi$ where $k = 1 \times 10^3 \mathrm{m^3 mol^{-1} s^{-1}}$. The values of parameters used in this system are similar to the reaction *Equation 2* in main text. 4 molecules of A and B each were simulated in a cylindrical arena of length $L = 500\,\mathrm{nm}$ and radius $r = 20\,\mathrm{nm}$. This arena was divided into $n$ equal subvolumes, each of length $h = L/n$. Diffusion of A and B was implemented as described *Equation 8* in where $D_A = D_B = D$. (A–E) Average of 20 trajectories of A vs. time for different combinations of $n$ and $D$. Error bars are standard-deviation. For fixed values of $n$ and non-zero values of $D$, the kinetics are indistinguishable from each other (panels A, B and C). This suggests that the results described in *Figure 4—figure supplement 1* are numerically correct. For fixed values of $D$, however, changing $n$ has a large impact on the dynamics of A (panels D,E) especially when there is less than 1 molecule in each voxel (compare plots where n = 1,4 vs n = 8,16). In all simulations presented here, we maintained at least one diffusing molecule in every voxel. (F) Time for A to reach zero ($t_0$) vs. $D$ and $n$.
DOI: https://doi.org/10.7554/eLife.41412.018

