## [Decision Letter]

Thank you for submitting your article "Subunit exchange enhances information retention by CaMKII in dendritic spines" for consideration by *eLife*. Your article has been reviewed by Gary Westbrook as the Senior Editor, a Reviewing Editor, and two reviewers. The reviewers have opted to remain anonymous. The reviewers have discussed the reviews with one another and the Senior Editor has drafted this decision to help you prepare a revised submission.

Summary:

The main goal of the work is to assess the impact on the stability of CaMKII phosphorylation states (a proposed switch for memory storage) of CaMKII subunit exchange, both between holoenzymes within the postsynaptic density (PSD) and of such exchange between the PSD and the spine itself. The question is a relevant one on an important topic and the simulations appear to be carried out with due care using a reasonable model of the biochemical pathways involved.

Essential revisions:

1) In many places the results are demonstrated by simulation, but little intuition is provided as to why these results occur. For example, any differences in behavior of prior models should be explained in terms of a causal link between which parameters are altered, or which additional reactions are in place, and how they impact the prior models. Some of the results (increased stability) are to be expected, as exchange between holoenzymes allows the majority to rescue the state of any individual that has by chance fluctuated away from the preferred state. Such intuition should be stated clearly.

2) One important, specific criticism is on the discussion of a dual decay rate of the CaMKII activation. Lack of fit to a single exponential does not provide evidence for a dual decay rate process, by which I think you mean a good fit to a double exponential. If you wish to make a positive statement about the type of decay process (rather than the currently justified negative statement-lack of fit to a single exponential) please fit with other functions and show the dual exponential is significantly better than other choices, using a criterion like AIC. Furthermore, in the Discussion section, you relate your non-exponential finding to a fractional power-law decay. However, I believe that both the dual-exponential and the fractional power-law functions decay more rapidly at short time-scales and more slowly at long time-scales than a single exponential. Yet your "best fit" exponential shows the error to be in the opposite direction. I think these discussions about power-laws and/or dual decay should be removed or reframed, unless a clear good fit to one of these other functions can be demonstrated.

3) The impact statement is not quite accurate. The authors have not shown, as far as I can see, that subunit exchange generates diverse timescales of information storage (and to be honest, it is unclear what "diverse timescales of information storage" really means). First, the decay of activity may not be a dual exponential, and second, it is unclear that subunit exchange is essential to produce the types of decay observed, given the large number of parameters in the system.

4) There were questions regarding the activation state of CaMKII and how some of the authors' parameters are defined. The most surprising result was that there is somehow an increase in "active" CaMKII (which one infers to mean Thr^286^ phosphorylated CaMKII) immediately in the presence of subunit exchange. When previously described, subunit exchange enhanced CaMKII activation at later times but did not affect early time points. To fully understand this, it is necessary for the authors to clearly define the following: (1) What is 'active' CaMKII, (2) what is the rate they are using for CaMKII activation (again assuming this to mean rate of Thr^286^phosphorylation) (3) What is the relationship between this initial rate of phosphorylation and subunit exchange. Some of this information is in the manuscript, but I think it needs to be more explicitly stated.

5) Figure 1C, where did these values/frequencies arise from?

6) Why was there no CaMKII activity in the absence of exchange (subsection “Subunit exchange facilitates the spread of CaMKII activity”)? Initial CaMKII activation should not be so affected by subunit exchange as this is actually faster than exchange itself.

7) The data in Figure 4B and 4C -> what is the explanation for the discrepancy in the affect on activity at 2 subunit exchange rates between 4B and 4C? (i.e. there is a large effect at 80 nM ca but zero effect at 120 nM ca).

8) Comments on the model: The assumption that both subunits within a vertical dimer are phosphorylated and dephosphorylated together is a large assumption – meaning that there is no direct evidence for this. I think this is OK to do, but it should be stated in the text as well.

9) Phosphorylation of CaMKII is mentioned many times, but it's never specified that it is Thr^286^phosphorylation. This should be made clear.

10) It is true that the rate of exchange in the cell is not known (subsection “Subunit exchange”). However, from in vitro experiments in Stratton et al., 2014; at ~4 μM CaMKII subunit concentration, the half-life of exchange is roughly 15 minutes. The rate was shown to increase at higher concentrations of CaMKII – so one could potentially extrapolate to the concentrations in the neuron (estimated to be roughly 100 μM in the spine).

11) Are newly synthesized holoenzymes considered in the model?

---

## [Author Response]

Essential revisions:1) In many places the results are demonstrated by simulation, but little intuition is provided as to why these results occur. For example, any differences in behavior of prior models should be explained in terms of a causal link between which parameters are altered, or which additional reactions are in place, and how they impact the prior models. Some of the results (increased stability) are to be expected, as exchange between holoenzymes allows the majority to rescue the state of any individual that has by chance fluctuated away from the preferred state. Such intuition should be stated clearly.

We have added context in many sections to provide some intuition behind the observations and claims being made, especially in the Results section related to the effect of the subunit exchange on the tolerance of PP1 and turnover.

Most results in this manuscript are due to subunit exchange. Since this is the first study to model subunit exchange in the context of CaMKII’s role in plasticity, no comparison could be made with previous models. However, we have expanded subsection”Model Validation”, to compare MZ model with our model (when subunit exchange and diffusion were disabled).

2) One important, specific criticism is on the discussion of a dual decay rate of the CaMKII activation. Lack of fit to a single exponential does not provide evidence for a dual decay rate process, by which I think you mean a good fit to a double exponential. If you wish to make a positive statement about the type of decay process (rather than the currently justified negative statement-lack of fit to a single exponential) please fit with other functions and show the dual exponential is significantly better than other choices, using a criterion like AIC. Furthermore, in the Discussion section, you relate your non-exponential finding to a fractional power-law decay. However, I believe that both the dual-exponential and the fractional power-law functions decay more rapidly at short time-scales and more slowly at long time-scales than a single exponential. Yet your "best fit" exponential shows the error to be in the opposite direction. I think these discussions about power-laws and/or dual decay should be removed or reframed, unless a clear good fit to one of these other functions can be demonstrated.

We apologize for the confusion caused by lack of details in the figure and text. We have expanded the relevant text and the figure caption to fix this. The key point, as we indicate in the text, is that there are both experimental and theoretical grounds for expecting that the process should be a double exponential. We therefore use this knowledge of the chemical system as a starting point and ask if our simulations are able to replicate these prior observations. From the revised text:

Subsection “Subunit exchange may account for the observed dual decay rate of CaMKII phosphorylation”: “Thus, if there are populations of clustered as well as non-clustered CaMKII in the spine, we expected that they would exhibit long and short time-courses activity decay, respectively. Therefore, a mixed population of clustered and non-clustered CaMKII should decay with two time-constants. Our simulations supported this prediction.

In Chang et al., (2017), the decay kinetics of CaMKII were obtained by curve fitting of experimental data… We used their estimate of P_fast_ and P_slow_ to construct a mixed population of slow and fast decaying CaMKII (Figure 6A, black), and simulated the decay kinetics of CaMKII for this system. We fit the resulting decay curve with a double-exponential function (Figure 6C, dashed red). The time-constants obtained (8.4 s, 86.2 s) matched well with experimentally estimated time-constants of (6.4 s ± 0.7, 92.6 s ± 50.7).”

Additionally, we have increased the number of stochastic simulation replicates (N=250 now, v/s 80 before) for better statistics. We have also increased the inter-stimulus gap to 1000 sec to better estimate the statistics of tail data. With these changes we have better resolution on the decay curves and can show that the exponential fit is accurate (see Figure 6B).

To summarise, the assertion of ‘dual-decay rate due to subunit exchange’ holds if in the spine cytosol there are two populations: (1) a population of clustered CaMKII in cytosol decaying slowly because subunit-exchange is effective over the small distances between holoenzymes; and (2) a fast-decaying population which is not clustered and therefore lacks subunit exchange.

3) The impact statement is not quite accurate. The authors have not shown, as far as I can see, that subunit exchange generates diverse timescales of information storage (and to be honest, it is unclear what "diverse timescales of information storage" really means). First, the decay of activity may not be a dual exponential, and second, it is unclear that subunit exchange is essential to produce the types of decay observed, given the large number of parameters in the system.

By diverse timescales, we meant the substantially different time-scales (separated by an order of magnitude or more) that emerge from CaMKII dynamics in the presence of subunit exchange. To make this explicit we have collected the time-scales from our study (in some cases corroborated by experiment) into Table 1 which we refer to in the Discussion section.

4) There were questions regarding the activation state of CaMKII and how some of the authors' parameters are defined. The most surprising result was that there is somehow an increase in "active" CaMKII (which one infers to mean Thr^286^ phosphorylated CaMKII) immediately in the presence of subunit exchange. When previously described, subunit exchange enhanced CaMKII activation at later times but did not affect early time points. To fully understand this, it is necessary for the authors to clearly define the following: (1) What is 'active' CaMKII, (2) what is the rate they are using for CaMKII activation (again assuming this to mean rate of Thr^286^phosphorylation) (3) What is the relationship between this initial rate of phosphorylation and subunit exchange. Some of this information is in the manuscript, but I think it needs to be more explicitly stated.

We have now indicated in the main text that by phosphorylation/activation we mean Thr^286^phosphorylation. We have also indicated values of parameters in the main text along with appropriate references/justifications. Additionally, we have also included typical values of parameters at resting Ca^++^ concentration of 100 nM.

Thank you for these suggestions to improve clarity. We have addressed each of these points:

1) What is 'active' CaMKII?

Subsection “Model validation”: “We define Active CaMKII as a holoenzyme (ring of 6 or 7 subunits) in which at least 2 subunits are phosphorylated at Thr^286^.”

2) What is the rate they are using for CaMKII activation (again assuming this to mean rate of Thr^286^phosphorylation)?

The rates are described by Equation 1. We have added following information in the main text, “k_1_ = 1.5 s^−1^ (Hanson et al., 1994), and k_H1_ = 0.7 μM (de Koninck and Schulman, 1998; Miller et al., 2005). At resting Ca^2+^ concentration of 100 nM, v_1_ = 1.27e-5 s^−1^ and v_2_ = 4.36e−3 s^-1^ (i.e., v_2_/v_1_ ≈ 343).

3) What is the relationship between this initial rate of phosphorylation and subunit exchange?

We now explicitly compare these rates in subsection “Phosphorylation and dephosphorylation of CaMKII ring”:

“The rate constant v_1_ above defines the initial rate of phosphorylation. Furthermore, addition of phosphorylated subunits can happen through subunit exchange (equation 3). We treat these as independent variables. The phosphorylation rates v_1_ and v_2_ are relatively well constrained by the experimental literature and are in the range of 1.25e^-5^/s and 4.36e^-3^/s respectively at basal calcium. The subunit exchange rates were estimated (Materials and methods) to be in the range of 1/s.

5) Figure 1C, where did these values/frequencies arise from?

The statistics of background calcium activity from “The Number of Glutamate Receptors Opened by Synaptic Stimulation in Single Hippocampal Spines”, [Nimchinsky et al., 2014]; in particular Figure 6 (adjusted to our spine volume). We assumed a uniform distribution (light blue histogram in A and B from the reference).

We have added subsection “Ca^2+^ background activity (*ϵ*)”:

“We assumed the resting Ca 2+ level in spine to be 80 nM (Berridge, 1998). […] This distribution is informed by Figure 2B,C from (Nimchinsky et al., 2004).”

We also refer to it in the legend of Figure 1C. We have also fixed a technical error in the description of the Ca^2+^ stimulus.

6) Why was there no CaMKII activity in the absence of exchange (subsection “Subunit exchange facilitates the spread of CaMKII activity”)? Initial CaMKII activation should not be so affected by subunit exchange as this is actually faster than exchange itself.

The reason for nearly zero CaMKII activity in the absence of exchange in Figure 4A is that the number of PP1 (N_PP1_) was chosen (from Figure 4D) so that there is no significant activity in the absence of subunit exchange at basal calcium level of 80nM+background noise. This was done since we wanted to quantify the effect of subunit exchange; this case acted as a baseline. We have explicitly indicated this in both figure caption and main text.

7) The data in Figure 4B and 4C -> what is the explanation for the discrepancy in the affect on activity at 2 subunit exchange rates between 4B and 4C? (i.e. there is a large effect at 80 nM ca but zero effect at 120 nM ca).

We assume that by “there is a large effect at 80 nM ca but zero effect at 120 nM ca” the reviewers meant the difference between red (D_sub_=1e^-8^) and blue (D_sub_=0) trajectories. We have added a more complete explanation in the Results section:

Subsection “Subunit exchange facilitates the spread of CaMKII activity”: “We ran simulations for 4 hours at basal calcium concentration [Ca^2+^]=80 nM+ε (where ε is the fluctuation in basal calcium levels, see Figure 1C), and without subunit exchange (i.e., D_sub_=0). We set NPP1 =15×N_CaMKII_ to make sure the system showed no significant CaMKII activity (Figure 4B, red curve). This served as the baseline to quantify the effect of subunit exchange. When we enabled subunit exchange by setting D_sub_ to a non-zero value, CaMKII activity rose to a maximum within 4 h even for a low value of D_sub_ =0.001 μm^2^/s (Figure 4C, black curves).”

Subsection “Subunit exchange facilitates the spread of CaMKII activity”: “The first step of CaMKII phosphorylation (Equation 1) is slow since it requires binding of two calcium/calmodulin complex (Ca^2+^/CaM) simultaneously (at basal [Ca^2+^] = 80 nM+ε, v1 = 1.27 × 10^−5^ s^−1^). However, subunit exchange can also phosphorylate a subunit in a holoenzyme by adding an available phosphorylated subunit to it (Equation 3). Note that a D_sub_ value as low as 0.001 μm^2^ s^−1^ is good enough for subunit exchange to be effective. With this value of D_sub_, it takes roughly 0.9 s for the subunit to reach another holoenzyme which is, on average, 30 nm away. Under these conditions the rate of picking up available active subunits (given in Equation 3) is faster than v_1_. Expectedly, for larger D_sub_ values (e.g., 0.001 and 0.1 μm^2^ s^−1^), subunit exchange becomes more effective (compare red and blue with the rest in Figure 4D).”

8) Comments on the model: The assumption that both subunits within a vertical dimer are phosphorylated and dephosphorylated together is a large assumption – meaning that there is no direct evidence for this. I think this is OK to do, but it should be stated in the text as well.

Yes. There is no direct evidence that this happens. However, this assumption helps us in treating the CaMKII rings as the proxy for holoenzyme. Without it, the complexity of model will increase 6-7 fold as there would be many more CaMKII states to be considered. This would have entailed a very significant simulation cost.

We state this assumption explicitly in subsection “Model validation”:

“We treat the CaMKII ring as a proxy for the CaMKII holoenzyme, which consists of two such rings stacked over each other”.

And in the Materials and methods section:

“We assume that vertical dimers are inserted and released together (Bhattacharyya et al., 2016). We also assume that both subunits of a vertical dimer phosphorylate and de-phosphorylate together. Under this assumption, we can treat the CaMKII ring as the proxy for the CaMKII holoenzyme and the subunit as the proxy for the CaMKII dimer. Without this assumption, the simulation cost of the increased complexity would be very significant.”

9) Phosphorylation of CaMKII is mentioned many times, but it's never specified that it is Thr^286^phosphorylation. This should be made clear.

We have made it clear now. We mention the site (Thr^286^) in each section whenever phosphorylation of CaMKII is mentioned for the first time.

*10) It is true that the rate of exchange in the cell is not known (subsection “Subunit exchange”). However, from* in vitro *experiments in Stratton et al., 2014; at ~4 μM CaMKII subunit concentration, the half-life of exchange is roughly 15 minutes. The rate was shown to increase at higher concentrations of CaMKII – so one could potentially extrapolate to the concentrations in the neuron (estimated to be roughly 100 μM in the spine).*

Thank you for this suggestion. We had overlooked this data. We have now incorporated an explicit model of this experiment into the analysis:

Subsection “Estimation of subunit exchange rate”: “Estimation of subunit exchange rate. To estimate reaction rates of Equation 3, we modeled the “single molecule assay” used in Stratton et al., (2014) […] within the physiologically relevant range.”

Fortunately, the values of subunit exchange rate which explains colocalization data “well” is close to the values which we have used in our simulations. A new figure is added to the manuscript (Figure 7) which summarises these results.

11) Are newly synthesized holoenzymes considered in the model?

Yes. Newly synthesised molecules (i.e. dodecamer or teradocamer with no subunit phosphorylated) replace any old molecules at a constant rate (v_t_). The equation 5 (v_t_=turnover rate) in the Materials and methods section models it, and Figure 3 summarises the result.